# Differentially Private Per-Instance Additive Noise Mechanism: A Game Theoretic Approach

## Abstract

Recently, the concept of per-instance differential privacy (pDP) has gained significant attention by virtue of its capability to assess the differential privacy (DP) of individual data instances within a dataset. Traditional additive mechanisms in the DP domain, which add identical noises to all data instances, often compromise the dataset's statistical utility to guarantee DP. A main obstacle in devising a per-instance additive noise mechanism stems from the interdependency of the additive noises: altering one data instance inadvertently affects the pDP of others. This intricate interdependency complicates the problem, making it resistant to straightforward solutions. To address this challenge, we propose a per-instance noise variance optimization (NVO) game, framed as a common interest sequential game. Our main contribution is that we show the Nash equilibrium (NE) points of this game inherently guarantee pDP for all data instances. We leverage the best response dynamics (BRD) algorithm to derive strategies for achieving the NE. To validate the efficacy of our approach, we evaluate the NVO game on various statistical metrics including regression experimental results. The source code to reproduce the results will be available soon.

## 1 Introduction

In the modern era, the rapid advancements in the field of machine learning have underscored the pivotal role of statistical datasets. This surge of data utilization inherently attracts an escalated focus on safeguarding against potential privacy risks. The principle of differential privacy (DP), a theoretical concept introduced by Dwork (2006), sheds light on the growing privacy concerns precipitated by the inclusion of individual data in the whole dataset. This approach strives to adeptly maneuver the delicate equilibrium between harnessing data for analytical advancements and ensuring the protection of individual privacy, thereby standing as a strong defense against the looming dangers of privacy breaches in today's digital landscape (Apple, 2017; Nguyên et al., 2016).

Conventional additive noise mechanisms that add noises with identical distributions to every data instance might not be optimal for preserving the dataset's informational utility. This one-size-fits-all approach fails to consider the privacy vulnerabilities of each data instance, which can vary significantly in terms of their distributional density. In essence, the less frequent a data point, the weaker its privacy assurance, a concept adeptly addressed in the per-instance DP (pDP) delineated by Wang (2019). That study drew attention to the varying levels of privacy protection in data instances when the same noise is added to each query output. However, it primarily focused on identifying and analyzing these issues, without offering concrete solutions. In contrast, our paper focuses on providing a concrete solution, a per-instance noise mechanism better preserving datasets' statistical utility. In the subsequent sections of this paper, we endeavor to address the following question:

> *When upholding DP for a dataset, is there a tailored noise distribution approach that can optimize its statistical utility on a per-instance basis?*

In response to this question, our objective is to introduce a per-instance additive noise mechanism grounded in the principles of pDP. To this end, we propose a noise variance optimization (NVO) game where the Nash equilibria ensure $\epsilon$-DP. Subsequently, we endeavor to obtain a Nash equilibrium (NE) strategy by executing a well-known game theory algorithm.

**Challenges**    A challenge on the horizon is optimizing the noise distribution to guarantee DP. One reason this direction has not been widely pursued is that ensuring pDP for a particular data instance is inherently dependent on the noise distribution of other data instances. Thus, altering the noise distribution for a data instance presents a tangible risk: certain instances might become non-compliant for pDP as a consequence of such modifications. Conventional additive noise mechanisms can guarantee mathematically well-proven assurance; however, when introducing non-identical noises, establishing these guarantees becomes more difficult. In summary, finding a balance between preserving the dataset's original statistical utility and ensuring $\epsilon$-pDP requires intricate adjustments to the noise distribution, a challenge amplified by the curse of interdependency.

**Contributions**    We introduce an innovative approach to optimize non-identical noise distribution tailored to specific data instances. Our salient contributions are as follows:

- We propose the NVO game designed to find suitable non-identical per-instance additive Laplace noises within a dataset. Within this game, every player (representing data instances) collaboratively/sequentially acts to guarantee $\epsilon$-pDP, all the while optimizing the utility of data statistics.
- We prove that an NE strategy in the NVO game ensures $\epsilon$-pDP across all data instances.
- We simulate the best response dynamics (BRD) algorithm as an example to obtain an NE strategy for the proposed NVO game. The proposed NVO game not only assures the same $\epsilon$-pDP as the commonly adopted Laplace mechanism but also demonstrates superiority in preserving statistical utility.

## 2    RELATED WORKS

**Additive mechanisms for DP**    Traditional additive noise mechanisms offer straightforward and mathematically well-proven methods to ensure DP. Efforts to refine these conventional methods abound: Geng & Viswanath (2014); Geng et al. (2015; 2019) have proposed additive staircase-like noise as a substitute for the Laplace distribution, aiming to optimize a given statistical utility function while guaranteeing $\epsilon$-DP. In addition, the IBM DP library showcases initiatives to boost additive noise mechanisms by clipping the randomized output within a pre-defined range (Holohan et al., 2019). In the realm of optimizing the noise distribution, Mironov (2017) has attempted to regulate additive noise to meet Rényi DP criteria. Parallel notions have been studied for similar concepts: sampling scenarios (Geumlek et al., 2017; Girgis et al., 2021) or deep learning (Wang et al., 2022; Zhu & Wang, 2020). While endeavors to modify the noise distributions are evident, previous studies have predominantly applied identical noises across all data. Such a method is not appropriate for guaranteeing tight DP while preserving statistical utility through per-instance non-identical noise.

**Relation to $(\epsilon, \delta)$-DP**    The pioneer of DP, Dwork & Roth (2014), defined $(\epsilon, \delta)$-DP, demonstrating that the Gaussian mechanism can achieve this definition. Since the advent of DPSGD (Abadi et al., 2016), there has been a surge of applications in machine learning (Ding et al., 2021; Moreau & Benkhelif, 2021; Truex et al., 2020). It is worth noting that our method can be easily adapted to the widely-known relaxation of DP, – namely $(\epsilon, \delta)$-DP, with the per-instance Gaussian mechanism.

**Game theory**    We present the NVO game, designed to determine the optimal variance of additive noises, taking cues from the game-theoretic perspective introduced by Neumann & Morgenstern (1944). Within the context of the NVO game, we show that the NE points of the game ensure $\epsilon$-DP. However, identifying this NE point brings its own set of complexities. To address this, we utilize established algorithms to reach the NE point (Taylor & Jonker, 1978; Zaman et al., 2018).

## 3    PRELIMINARY

In this section, we introduce the preliminary concepts underpinning this paper. For brevity, we use scalar-form data instances in the remainder of this paper. We note that this work can be easily extended to vector-form data instances. We begin with the definition of $\epsilon$-pDP, a concept tailored to assess per-instance privacy loss for a *dataset*.

**Definition 3.1** ($\epsilon$-pDP (Wang, 2019)). *A randomized mechanism, denoted by $\mathcal{M}$, has a range $\mathcal{R}(\mathcal{M})$. For a fixed dataset $\mathcal{Z}$ and a fixed data instance $z \in \mathcal{Z}$, the mechanism $\mathcal{M}$ meets $\epsilon$-pDP, if the following condition holds:*

$$\left| \ln \frac{\Pr[\mathcal{M}(\mathcal{Z}) \in S]}{\Pr[\mathcal{M}(\mathcal{Z} \setminus \{z\}) \in S]} \right| \le \epsilon, \ \forall S \subseteq \text{Range}(\mathcal{M}). \tag{1}$$

We note that a randomized mechanism, $\mathcal{M}$, qualifies as an $\epsilon$-DP mechanism if it maintains $\epsilon$-pDP for all data instances and datasets. For guaranteeing either $\epsilon$-DP or $\epsilon$-pDP, the most recognized method is the Laplace mechanism, which we detail in the subsequent definition.

**Definition 3.2** (Laplace mechanism). *Given any query function $f : \mathcal{X} \to \mathbb{R}$ with $\ell_1$ sensitivity of $\Delta f \in \mathbb{R}$, the Laplace mechanism is defined as:*

$$\mathcal{M}_{\text{L}}(x, f(\cdot), \epsilon) = f(x) + y, \tag{2}$$

*where $y$ is a random number drawn from $\text{Lap}(\Delta f / \epsilon)$ and $\mathcal{X}$ denotes the domain of variable $x$.*

Given that the Laplace mechanism guarantees $\epsilon$-DP by addressing the worst-case scenario, it offers a chance to enhance dataset utility by employing a customized per-instance noise distribution. In this study, we focus on the random sampling query, as outlined subsequently.

**Definition 3.3** (Random sampling query). *Given numeric dataset $\mathcal{D}$, the output of the random sampling query $q$ is a random variable following the probability distribution of a dataset, i.e.,*

$$q(\mathcal{D}) \sim \Pr(\mathcal{D}). \tag{3}$$

This query encapsulates the statistical attributes of the dataset, given that it directly fetches an instance from it.

**Remark 3.1** (Extensibility of random sampling query). *The random sampling query is a fundamental query that encompasses all possible statistical queries. This is because the random sampling query can capture the statistical distribution of a dataset. Thus, from the post-processing theorem, achieving pDP/DP for random sampling queries can guarantee pDP/DP for all statistical queries.*

The Nash equilibrium, a foundational concept in a game theory introduced by Nash (1951), denotes the ideal state of a game where every player makes their optimal decision based on the choices of their counterparts as below.

**Definition 3.4** (Nash equilibrium). *A Nash Equilibrium is a profile of strategies $(s_i^*, s_{-i}^*)$, such that each player's strategy is an optimal response to the other players' strategies: $\Pi_i(s_i^*, s_{-i}^*) \ge \Pi_i(s_i, s_{-i}^*)$, $\forall i$ where $s_{-i}$ is the strategy profile of all players except for player $i$ and $\Pi_i(s)$ is a payoff function.*

To solidify DP assurances, we frame the challenge of noise addition for data utility maximization as a game, aiming to reach an NE point.

## 4 NOISE VARIANCE OPTIMIZATION GAME

In this section, our focus is to design a sequential/cooperative game that applies per-instance Laplace noises to the target dataset, ensuring $\epsilon$-pDP. We denote the target dataset by $\mathcal{D}$, and its data instances are represented by $d \in \mathbb{R}$. We assume that the target dataset consists of real-valued data instances, *e.g.*, a regression dataset. The problem we aim to solve using game theory can be defined as a constrained optimization problem:

$$\min U(\mathcal{D}, \mathcal{M}(\mathcal{D})) \quad \text{s.t.} \left| \ln \frac{\Pr[\mathcal{M}(\mathcal{D}) \in S]}{\Pr[\mathcal{M}(\mathcal{D} \setminus \{d\}) \in S]} \right| \le \epsilon, \forall d \in \mathcal{D}, \forall S \in \text{Range}(\mathcal{M}), \tag{4}$$

where the function $U$ represents an arbitrary utility function of the original dataset $\mathcal{D}$ and randomized dataset $\mathcal{M}(\mathcal{D})$. In our work, KL divergence is used as our utility function.

An illustration of the proposed game design, including the preprocessing step, is depicted in Fig. 1. Detailed explanations corresponding to this figure will be covered in the subsequent portions of this section.

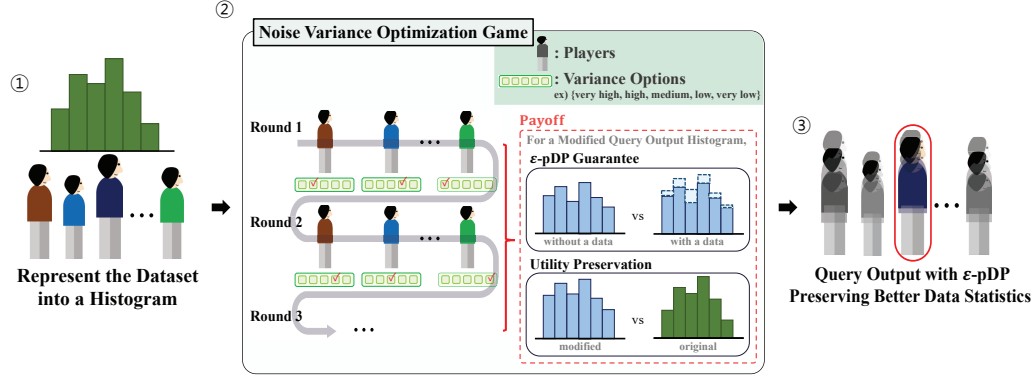

Figure 1: The procedure entails identifying the best mix of noise variances for the query outputs of each data instance. In Step 1, we represent the dataset as a histogram through normalization and categorization. In Step 2, we aim to find an NE point for the NVO game, where multiple players iteratively update their variance parameters to ensure $\epsilon$-pDP while preserving statistical utility. Once the prime set of noise variances is determined, Step 3 allows us to formulate queries that assure $\epsilon$-pDP by executing a random sampling query.

## 4.1 STEP 1: A HISTOGRAM REPRESENTATION OF THE DATASET

**Normalization** In the context of the NVO game, assessing the probability density function of the mechanism's output for every single point across all data instances is computationally daunting. Also, as pointed out by Abadi et al. (2016), the most accurate way for gauging privacy loss is manual integration within designated intervals rather than relying on theoretical boundaries. For the manual integration, we normalize the dataset into the $[0, 1]$ range by min-max normalization. We conservatively opt to set our integration's target range to encompass $p$-percentile of the Laplace mechanism with the target $\epsilon$, which is defined by $(-(\Delta q/\epsilon)\ln(2-2p), (\Delta q/\epsilon)\ln(2-2p))$ if $p > 0.5$[1], where $\Delta q$ denotes the sensitivity of random sampling query $q$. For brevity, we denote $(\Delta q/\epsilon)\ln(2-2p)$ as $\Delta_{(\epsilon,p)}$. Then, the min-max linear normalization is executed in the interval $[d_{\min} - \Delta_{(\epsilon,p)}, d_{\max} + \Delta_{(\epsilon,p)}]$, where $d_{\min} = \min_i d_i$ and $d_{\max} = \max_i d_i$.

**Categorization** After normalization, the continuous nature of the domain $S$ poses a unique challenge, differing from that in $\epsilon$-DP. In this context, it is necessary to confirm the $\epsilon$-pDP condition for each instance and each point within $S$, an endeavor impractical to accomplish in polynomial time. To address this issue, we make $K$ non-overlapping intervals into the range of the dataset and allocate each data instance $d \in \mathcal{D}$ into categories based on their corresponding intervals. We set the $K$ uniformly divided non-overlapping intervals as follows: the $i$-th interval is $(\frac{i}{K}, \frac{i+1}{K})$, where the representative value of each interval bin is the midpoint of the interval. The set of the representative values is denoted as $\mathcal{K}$. Through data categorization, the dataset can be represented in the form of a histogram. An advantage of segmenting data instances into distinct bins is that certain data instances can inherently ensure a non-infinite $\epsilon$, whereas the original continuous dataset cannot[2].

## 4.2 STEP 2: A GAME OF NOISE VARIANCE OPTIMIZATION

Here, we introduce the NVO game following data preprocessing. In this game, every data instance possesses data values within the range $[0, 1]$. The classes within our proposed NVO game include: i) *sequential game*, ii) *fully-cooperative game*, iii) *potential game*, and iv) *common interest game*.

---

[1] In our experiments, the value of $p$ is set to be 0.9.

[2] Once categorized, if a category contains three data points, the inherent $\epsilon$-pDP assurance for these instances is given by $\log 3/2 \approx 0.4$.

### 4.2.1 DEFINITIONS OF PLAYERS, STRATEGIES, AND PAYOFFS

**Players** In this study, each data instance, acting as a player, collaboratively and sequentially participates in the NVO game. The goal of them is to establish $\epsilon$-pDP (their respective payoffs) by designing their strategies (variance optimization), simultaneously maximizing the dataset's statistical utility. We note that the player of the NVO game is represented by $I = \{1, 2, \ldots, |\mathcal{D}|\}$.

**Strategies** With data instances cast as players, the strategy for player $i$ is defined by the additive noise applied to data $d_i$. Denoting the variance of additive noises to the $i$-th data instance $d_i$ as $b_i$, the action of the player $i$ is written by $b_i$. That is, from the random sampling query in Definition 3.3, the per-instance Laplace mechanism is articulated as

$$\mathcal{M}(d_i) = d_i + y_i, \tag{5}$$

where $y_i$ is a random variable drawn form $\text{Lap}(b_i)$. Typically, games featuring strategy sets of uncountable infinity might not always present an NE solution. As a result, we confine these strategy sets to a discrete domain. In other words, the added variance is chosen from a discrete set $\mathcal{V} = \{v_1, v_2, \ldots, v_n\}$.

**Payoffs** In the context of the NVO game, the payoff should induce the player to primarily uphold $\epsilon$-pDP and secondarily preserve the dataset's statistical utility. In this domain, there is a trade-off between data statistics and privacy. Emphasizing robust privacy can reduce query output quality, while maximizing utility might compromise privacy. The goal is to optimize utility without violating the $\epsilon$-pDP constraint, achievable by carefully adjusting noise variance on query results. With this in mind, we articulate the overall payoff $P(\mathcal{M}, \mathcal{D})$ as a composite of two objectives: privacy assurance $P_E(\mathcal{M}, \mathcal{D})$ and utility preservation $P_U(\mathcal{M}, \mathcal{D})$. We note that the players in the proposed NVO game cooperate to benefit from the shared payoffs. Simply, the NVO game is characterized as a kind of common interest game.

### 4.2.2 PRIVACY ASSURANCE PAYOFF

The payoff related to privacy assurance, denoted as $P_E$, functions as an indicator of how effectively $\epsilon$-pDP is met for a dataset, viewed through the lens of pDP. Let us define $p_{\epsilon,i}$ as an indicator for representing whether the $i$-th data instance's pDP is satisfied or not, *i.e.,*

$$p_{\epsilon,i}(\mathcal{M}, \mathcal{D}) = \begin{cases} 1, & \text{if } d \in \mathcal{D} \text{ satisfies the } \epsilon\text{-pDP condition in Definition 3.1,} \\ 0, & \text{otherwise.} \end{cases} \tag{6}$$

Then, the privacy assurance payoff $P_E$ is defined as the number of data instances satisfying the $\epsilon$-pDP condition as

$$P_E(\mathcal{M}, \mathcal{D}) = \sum_{i=1}^{|\mathcal{D}|} p_{\epsilon,i}(\mathcal{M}, \mathcal{D}). \tag{7}$$

Having established the privacy assurance payoff, we are presented with two subsequent questions: one concerning utility preservation and the other about ensuring privacy assurance at the NE point.

- *Q1: How do we determine the utility preservation payoff?*
- *Q2: Does the NVO game truly ensure $\epsilon$-pDP for all data instances using the privacy assurance payoff as outlined in Equation 7?*

### 4.2.3 UTILITY PRESERVATION PAYOFF *(Answer to Q1)*

In response to *Q1*, we formulate the utility preservation payoff $P_U(\mathcal{M}, \mathcal{D})$ to measure the statistical difference between the original dataset $\mathcal{D}$ and the randomized dataset $\mathcal{M}(\mathcal{D})$. Here, a higher value indicates a smaller difference. Furthermore, we ensure the utility preservation payoff does not compromise the assurance of $\epsilon$-pDP by scaling the targeted utility function $U$ into the range $[0, 1]$. It is pertinent to mention that a variety of statistical utilities can be adopted as the utility function, encompassing metrics like $n$-th order momentum, Kullback-Leibler (KL) divergence, and Jensen-Shannon (JS) divergence, among others.

In this paper, for example, we set the utility function by using KL divergence, i.e., $U(q(\mathcal{D})||q(\mathcal{M}(\mathcal{D}))) = D_{\mathrm{KL}}(q(\mathcal{D})||q(\mathcal{M}(\mathcal{D}))),$ as in the following remark.

**Remark 4.1** (Examples of the utility function). *For a dataset $\mathcal{D}$, the output probability distribution $q(\mathcal{D})$ of a query $q$, and a randomized function $\mathcal{M}$, the utility preservation payoff is defined as*

$$P_{\mathrm{U}}(\mathcal{M}, \mathcal{D}) = 1 - \frac{D_{\mathrm{KL}}(q(\mathcal{D})||q(\mathcal{M}(\mathcal{D})))}{\log(K)} \in [0, 1], \tag{8}$$

*where the utility function $D_{\mathrm{KL}}(q(\mathcal{D})||q(\mathcal{M}(\mathcal{D})))$ is bounded in $[0, \log(K)]$. The minus sign is used since the KL-divergence is a measure of information-theoretic distance between two probability distributions, where a smaller value indicates greater similarity between the distributions. The bound can be obtained by the fact that $\log(K) \geq \log \frac{q(\mathcal{D})}{q(\mathcal{M}(\mathcal{D}))}$.*

### 4.2.4 GUARANTEE OF THE $\epsilon$-DP CONDITION *(Answer to Q2)*

In addressing the previously posed question, *Q2: "Does the NVO game truly ensure $\epsilon$-pDP for all data instances using the privacy assurance payoff as outlined in Equation 7?"*, we set forth a proof demonstrating that the NE strategy for the proposed NVO game can consistently ensure $\epsilon$-pDP for a dataset.

**Theorem 4.1.** *Let us define the minimum variance in the set of possible action $\mathcal{V}$ as $b_{\min} \neq 0$. Then, $\epsilon$-pDP for all data instances upholds if the following condition is satisfied:*

$$b_{\min} \geq \frac{1}{\log\left(1 + (|\mathcal{D}| - 1)(\exp(\epsilon) - 1)\right)}. \tag{9}$$

In Theorem 4.1, we show that an NE point for the NVO game guarantees the $\epsilon$-pDP for all $d \in \mathcal{D}$ if the condition in Equation 9 holds. In the theorem, there always exists a value $b_{\min}$ that makes the NE point of the NVO game ensure the $\epsilon$-pDP for all $\epsilon \geq 0$.

**Remark 4.2** (Intuition of Theorem 4.1). *In Equation 9, if there are more data instances in the dataset, the influence of the individual data point diminishes, thereby allowing us to guarantee $\epsilon$-pDP easily. That is, if the value of $|\mathcal{D}|$ increases, we can guarantee $\epsilon$-pDP with smaller variance noise. On the other hand, if $\epsilon$ decreases to zero, query output with and without a data point should be statistically the same. Thus, the variance of the added noise becomes infinite, resembling a uniform query output distribution.*

## 5 ALGORITHM FINDING THE NASH EQUILIBRIUM OF THE NVO GAME.

In this section, we delve into an algorithm designed to secure an NE strategy within the framework of the NVO game. We begin by showcasing the BRD algorithm, adapted specifically for this game.

**BRD algorithm** The BRD algorithm is a concept in game theory where players, taking into account the current strategies of their opponents, opt for their most favorable response. During this iterative process, players sequentially decide on their best actions, which is presented in Alg. 1. The choice of values within the variance space can be tailored to encompass all the possible noise variance values. As the cardinality of the variance space $\mathcal{V}$ expands, the algorithm's performance improves, but there is a significant increase in computational complexity. Therefore, it is crucial to define the variance set $\mathcal{V}$ considering the trade-off between computational complexity and utility.

**Common interest game & potential game** In a common interest game, participants share a unified payoff. A player's strategy change directly impacts both the potential function and their own payoff, classifying it inherently as a potential game. Essentially, every data point acts as a cooperative player aiming for a joint goal. By crafting a shared payoff to maximize and iteratively selecting the optimal noise for each data instance's output, achieving an NE is feasible.

**Convergence of BRD toward NE** As shown by Boucher (2017), the BRD algorithm always converges into an NE point, if the target game belongs to one of the following games: potential games,

---

**Algorithm 1** Best response dynamics (BRD) for NVO game

---

**Input** dataset $\mathcal{D} = \{d_i | i = 1, ..., m\}$, variance space $\mathcal{V} = \{v_i | i = 1, ..., n\}$, target epsilon value $\epsilon$

    $i \leftarrow 1$ and $p^* \leftarrow 0$                                     ▷ Initialize data index and the best payoff

    $V[l] \leftarrow$ randomly initiates from $v \in \mathcal{V}$, for $l = 1, \ldots, |\mathcal{D}|$      ▷ Initialize the best variance set

    **while** $p^*$ converges over the dataset **do**

        PAYOFF$[l] \leftarrow 0$, for $l = 1, \ldots, |\mathcal{V}|$

        **for** $j \leftarrow 1$ to $|\mathcal{V}|$ **do**             ▷ Explore and store payoffs for all variance options

            V_temp $\leftarrow$ V

            V_temp$[i] \leftarrow v_j$

            PAYOFF$[j] \leftarrow$ GET_PAYOFF$(\mathcal{D},$ V_temp$, \epsilon)$

        **end for**

        $p^* \leftarrow \max$ PAYOFF and $j^* \leftarrow \text{argmax}$ PAYOFF

        V$[i] \leftarrow v_{j^*}$                              ▷ get the best variance for a current element

        $i \leftarrow (i + 1) \bmod |\mathcal{D}|$

    **end while**

    **return** V                                                  ▷ Nash equilibrium

---

*GET_PAYOFF() is a function of the proposed payoff by summing up Equations 7 and 8.

weakly acyclic games, aggregative games, and quasi-acyclic games. As noted above, the NVO game is a potential game; thus, the BRD algorithm can obtain an NE point of the NVO game.

From the proof of Theorem 4.1 in the Appendix A, when every variance option exceeds $b_{\min}$, there always exists a choice that consistently increases one pDP assurance at each round. Hence, guaranteeing $\epsilon$-pDP for all data instances is feasible after $|\mathcal{D}|$ rounds. Intuitively, as players opt for their best responses, either sequentially or simultaneously, the potential function's value rises, eventually peaking. The strategy at this peak is the game's NE.

## 6 EXPERIMENTS

In this section, we evaluate the NE strategy of the NVO game. Our primary focus is to observe if the proposed NVO game has a superior dataset's statistical utility than the conventional Laplace mechanism while maintaining the same level of $\epsilon$-pDP/DP.

**Dataset** To assess this, we conduct simulations on two publicly available datasets: 1) NBA player dataset[3] and 2) personal income dataset[4]. In the main manuscript, we only show the results on the NBA player dataset. The results for the personal income dataset can be found in Appendix E. In the NBA players dataset, we employ 1,307 data instances with the tuple of (height and weight) for players who joined five teams from 1963 to 2021: *Atlanta Hawks*, *Boston Celtics*, *Charlotte Hornets*, *Chicago Bulls*, and *Cleveland Cavaliers*. The results with a more complex dataset are available in Appendix F.

**Experimental detail** In our experiments, we configure the target $\epsilon$ values in $\{1, 2, 4, 8\}$. After normalization and discretization, the height and weight values belong to 101 categories, *i.e.*, $K = 101$. For the action of the players, variance set $\mathcal{V}$ is defined by $\{3 \times \Delta q/\epsilon, 2 \times \Delta q/\epsilon, \Delta q/\epsilon, 0.33 \times \Delta q/\epsilon, 0.2 \times \Delta q/\epsilon\}$. From Theorem 4.1, $\epsilon$-pDP for the smallest $\epsilon$ is achievable with the configured variance set $\mathcal{V}$, since $b_{\min} \approx 0.129$. That is, the NE points in the proposed NVO game ensure $\epsilon$-pDP. For comparison, we additionally implemented an approximated enumeration (AE) algorithm based on the genetic algorithm, with excessive generations. For more details, please refer to Appendix B.

### 6.1 ANALYSIS 1: STATISTICAL UTILITY

Here, we first analyze the dataset's statistical utility after executing randomized mechanisms for the height feature of the NBA player dataset. In Fig. 2, we compare the probability distribution of the

---

[3]https://www.kaggle.com/datasets/justinas/nba-players-data
[4]https://www.kaggle.com/datasets/mastmustu/income

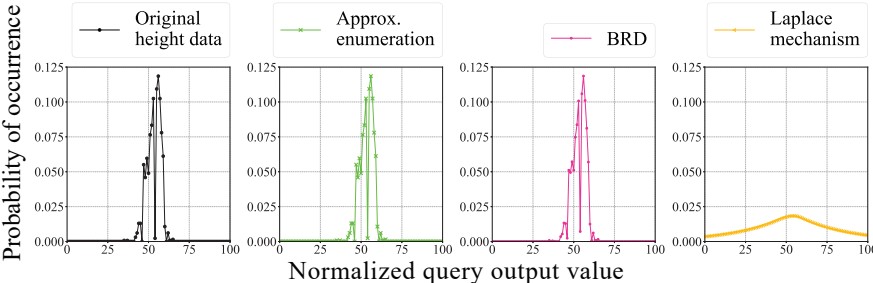

Figure 2: Comparison of query output probability distributions for the height data with each algorithm for $\epsilon = 1$. The $x$-axis represents the index of the categorization bins from 0 to 100 ($K = 101$).

Table 1: Each algorithm's average computation time, KL divergence, L1 loss of SD, Jaccard index with a threshold 0.001, and cosine similarity are evaluated for the height data. The modified query output distributions for all algorithms satisfy $\epsilon$-pDP.

| Algorithm | Target $\epsilon$ | Comp. time ↓ (minutes) | KL divergence ↓ | L1 SD loss ↓ | Jaccard index ↑ (threshold=0.001) | Cos. similarity ↑ |
|---|---|---|---|---|---|---|
| BRD | $\epsilon$=1 | **4** | **0.0066** | **0.0049** | **0.9523** | 0.9992 |
| | $\epsilon$=2 | 5 | **0.0045** | 0.0084 | 0.9523 | 0.9997 |
| Approx. enum. | $\epsilon$=1 | 392 | 0.0176 | 0.0058 | **0.9523** | **0.9998** |
| | $\epsilon$=2 | 354 | 0.0047 | **0.0080** | **1.0000** | **0.9999** |
| Laplace mechanism (baseline) | $\epsilon$=1 | - | 1.3991 | 0.0261 | 0.1980 | 0.5656 |
| | $\epsilon$=2 | | 0.9480 | 0.0247 | 0.2631 | 0.7040 |

*Best: **bold**, second-best: underline.

randomized mechanisms' output. As shown in the figure, the proposed NVO game (BRD and AE) has more similar shapes of distribution to the original one than the conventional Laplace mechanism.

In Table. 1, we quantitatively measure various statistical utility functions: 1) KL divergence, 2) L1 loss of standard deviation (SD), 3) Jaccard index, and 4) cosine similarity of the distribution. For more details on the metrics, please refer to Appendix C. In the table, the NVO game-related algorithms have superior statistical utility than the Laplace mechanism at 99.53%. Despite its superior performance, the AE algorithm requires much longer computation time than the BRD algorithm.

In Fig. 3, the SD of the added noise in each categorization bin is depicted. As we can observe, compared to the conventional Laplace mechanism, the BRD and AE algorithms add relatively small variance noises, thereby achieving superior statistical utility.

## 6.2 ANALYSIS 2: REGRESSION TASK

In order to evaluate the practical usefulness of the randomized mechanisms, we conduct a simulation of a regression task to estimate the weight feature from the height feature of the NBA player dataset. For the regression task, we configure a multi-layer neural network, which consists of three layers with ten parameters activated by the Rectified Linear Unit (ReLU) function. There are three different neural networks trained with 1) the original dataset, 2) a randomized dataset with the NVO game, and 3) a randomized dataset with the conventional Laplace mechanism. In Fig. 4, we show the scatter diagram of the preprocessed original dataset, and the height-weight regression curve of the datasets. Compared to the conventional Laplace mechanism, the regression curve of the NVO game is more similar to that of the original data. Quantitatively, as shown in Table 2, even with the case of low epsilon, 1-DP, the average RMSE of the BRD algorithm is apart from only 8.6% from that of the original dataset. For 8-DP, the average RMSE for the prediction of the BRD algorithm and original regression are almost the same.

Table 2: The average RMSE loss for regression task for the entire dataset, where the samples were generated using the noise associated with the pDP/DP algorithms.

| Algorithm | Average RMSE | | | |
|---|---|---|---|---|
| | $\epsilon = 1$ | $\epsilon = 2$ | $\epsilon = 4$ | $\epsilon = 8$ |
| Original data (reference) | 0.0218 | | | |
| NVO game (BRD) | **0.0227** | **0.0221** | **0.0219** | **0.0218** |
| Laplace mechanism | 0.0444 | 0.0380 | 0.0335 | 0.0272 |

*Best: **bold**.

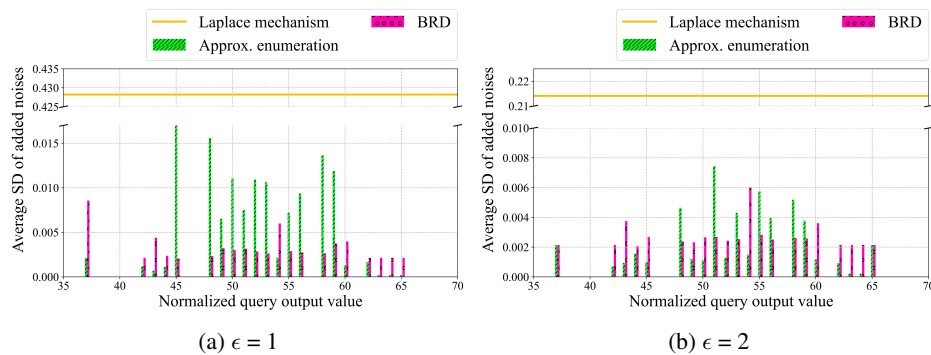

Figure 3: Distributions of average noise standard deviation for the height dataset for $\epsilon = 1$ and 2.

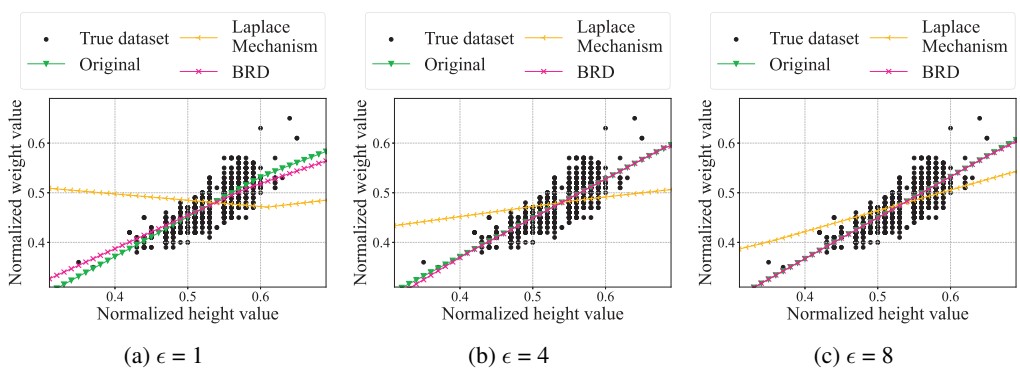

Figure 4: Linear regression result of 500 sampled data for each algorithm for $\epsilon = 1$, 4, and 8.

## 7 DISCUSSION

**Summary** We introduce the NVO game, accounting for the optimization of the per-instance Laplace mechanism. We directly tackle the noise variance optimization problem for $\epsilon$-pDP, aiming to maximize a pre-defined statistical utility function. We prove that the NE point of the NVO game guarantees $\epsilon$-pDP for all data instances, in which there have been a fluent of game theoretic methods to achieve this. In experiments, we demonstrate that the proposed method dramatically outperforms the conventional Laplace mechanism in various statistical utility metrics.

**Extensibility to other queries** As discussed in Remark 3.1, the random sampling query is a fundamental query that can encompass all statistical queries targeted by differential privacy. Therefore, this study represents a universal framework applicable to the full spectrum of statistical queries.

**Limitations** The target distribution used in our work is based on Laplace distribution. Also, in our proposed NVO game, the original dataset is categorized into several bins to ensure low computational complexity. More importantly, the choice of the noise variances is limited to a discrete set, even though we have shown the convergence to the $\epsilon$-pDP. In addition, the size of datasets is getting extremely large, and the payoff computation exponentially increases $|\mathcal{D}|$.

**Further research directions** We note that future work should explore the optimization of noise variances within a discrete space and investigate alternative noise distributions beyond the Laplace noises. Furthermore, the broader applicability of the NVO games concept across various domains remains an exciting avenue to explore, such as classification datasets.

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

## A  PROOF OF THEOREM 4.1

**Theorem 4.1.** *Let us define the minimum variance in the set of possible action $\mathcal{V}$ as $b_{\min} \neq 0$. Then, $\epsilon$-DP upholds if the following condition is satisfied:*

$$b_{\min} \geq \frac{1}{\log\left(1 + (|\mathcal{D}| - 1)(\exp(\epsilon) - 1)/K\right)}. \tag{10}$$

For simplicity of the proof, we tackle the situation for the scalar dataset case of the NVO game, *i.e.,* $d = 1$. We note that the proof can be extended to the vector version by considering each element separately.

## A.1 NOTATIONS

In this proof, we use the following notations:

- $b_i$: Action of the player $i$, the variance of noise added to data $d_i$.
- $(b_i, b_{-i})$: Set of each players' strategy.
- $b^* = (b_i^*, b_{-i}^*)$: An NE point of the NVO game.
- $b_{\min} = \arg\min_{b \in \mathcal{V}} b$.
- $\mathcal{K}$: The set of possible query output values, s.t. $\mathcal{K} \subset [0, 1]$.
- $m_{i,x}$: The overall probability mass added to $x \in \mathcal{K}$ by the noise assigned to the $i$-th data instance.
- $M_{-i,x} = \sum_{j \in [|\mathcal{D}|] \setminus \{i\}} m_{j,x}$
- $v_{\min}, v_{\max}$: The minimum and maximum values of the probability mass $m_{i,x}$ added to $x \in \mathcal{K}$ by additive noise to the $i$-th data instance.
- $\Pi_i(b_i, b_{-i})$: The $i$-th player's payoff for the strategy $(b_i, b_{-i})$.
- $P_{i,\mathrm{E}}, P_{i,\mathrm{U}}$: The $i$-th player's $P_\mathrm{E}$ and $P_\mathrm{U}$ for the strategy.
- $\Delta P_{i,\mathrm{E}}, \Delta P_{i,\mathrm{U}}$: The change of $P_\mathrm{E}$ and $P_\mathrm{U}$ values for the $i$-th player, s.t. $\Delta P_{i,\mathrm{E}} = P_{i,\mathrm{E}} - P_{i-1,\mathrm{E}}$ and $\Delta P_{i,\mathrm{U}} = P_{i,\mathrm{U}} - P_{i-1,\mathrm{U}}$.

## A.2 ASSUMPTION

Let us assume that we implement the NVO games with a continuous variance space $\mathcal{V} = [b_{\min}, \infty)$ for $b_{\min} \neq 0$ and the set of possible query output values $\mathcal{X} = [0, 1]$. We do not add noise with a probability of occurring outside the target range of the integration (*i.e.,* $[0, 1]$); thus, the probability density function of the Laplace noise is normalized as in Equation 14.

## A.3 PROOF

**The worst case to ensure $\epsilon$-pDP for an data instance** For the proof of the theorem, we start with the worst case of the $\epsilon$-pDP of an arbitrary data instance. In order to satisfy $\epsilon$-pDP for an element $d_i$, the following condition should be satisfied:

$$\max_x \frac{m_{i,x} + M_{-i,x}}{M_{-i,x} \cdot \frac{|\mathcal{D}|}{|\mathcal{D}-1|}} < \max_x \frac{m_{i,x} + M_{-i,x}}{M_{-i,x}} \leq \exp(\epsilon) \tag{11}$$

$$\Rightarrow \max_x \frac{m_{i,x} + M_{-i,x}}{M_{-i,x}} \leq \max_x \frac{m_{i,x} + \min M_{-i,x}}{\min M_{-i,x}} = \max_x \frac{m_{i,x} + (|\mathcal{D}| - 1)v_{\min}}{(|\mathcal{D}| - 1)v_{\min}} \tag{12}$$

$$= \frac{m_{i,q(d_i)} + (|\mathcal{D}| - 1)v_{\min}}{(|\mathcal{D}| - 1)v_{\min}} \leq \exp(\epsilon), \tag{13}$$

where Equation 11 is initialized from the definition of $\epsilon$-pDP.

**Find the $v_{\min}$** The minimum value of the $m_{i,x}$, represented by $v_{\min}$ is obtained by

$$v_{\min} = \min_{i,x} m_{i,x} = \min_{\substack{\mu,x \in [0,1] \\ b \geq b_{\min}}} \frac{\frac{1}{2b} \exp(-\frac{|x-\mu|}{b})}{\int_0^1 \frac{1}{2b} \exp(-\frac{|t-\mu|}{b}) dt} \tag{14}$$

$$= \min_{\substack{\mu,x \in [0,1] \\ b \geq b_{\min}}} \frac{\frac{1}{2b} \exp(-\frac{|x-\mu|}{b})}{1 - \frac{1}{2} \exp(\frac{\mu-1}{b}) - \frac{1}{2} \exp(\frac{-\mu}{b})} = \min_{\substack{\mu,x \in [0,1] \\ b \geq b_{\min}}} V(\mu, b, x). \tag{15}$$

In Equation 14, the definition of $v_{\min}$ is rewritten by the Laplace distribution $f(x|\mu, b) = \frac{1}{2b}\exp(-\frac{|x-\mu|}{b})$. For brevity, in Equation 15, we newly define a function $V(\mu, b, x)$.

Then, our focus is to find a value of $\mu$ for the $v_{\min}$, and check the critical points with following conditions:

$$\frac{\partial V}{\partial \mu} = 0 \tag{16}$$

$$\Rightarrow \frac{1}{2b^2}\exp(\frac{\mu-x}{b})\left[1-\frac{1}{2}\exp(\frac{\mu-1}{b})+\frac{1}{2}\exp(\frac{-\mu}{b})\right]-\frac{1}{2b}\exp(\frac{\mu-x}{b})\left[-\frac{1}{2b}\exp(\frac{\mu-1}{b})+\frac{1}{2b}\exp(\frac{-\mu}{b})\right] \tag{17}$$

$$= \frac{1}{2b^2} = 0, \tag{18}$$

where Equation 17 holds because of the Laplace distribution's symmetry, thereby making us to consider $x \geq \mu$. Then, from the result of Equation 18, we confirm that there is no critical point that makes $\partial V/\partial \mu = 0$. Also, when $x \geq \mu$, we confirm that the sign of $\partial V/\partial \mu$ is always positive; thus, the minimizer $\mu$ of the function $V(\mu, b, x)$ is zero as follows:

$$\text{sign}\left(\frac{\partial V}{\partial \mu}\right) = \text{sign}\left(\frac{1}{2b}\right) = \frac{1}{2b^2} \geq \frac{1}{2b_{\min}^2} > 0 \tag{19}$$

$$\Rightarrow \arg\min_\mu V(\mu, b, x) = 0. \tag{20}$$

Then, by substituting $\mu = 0$ into $V(\mu, b, x)$, we confirm that the minimizer $x$ of the function is one as follows:

$$\frac{\partial V}{\partial x} = \frac{-\frac{1}{b^2}\exp(-\frac{x}{b})}{1-\exp(-\frac{1}{b})} < 0 \Rightarrow \arg\min_{x\in[0,1]} V(0, b, x) = 1. \tag{21}$$

Up to here, we obtained the minimizers $\mu = 0$ and $x = 1$. By substituting the minimizers, we can obtain the minimizer $b$ as

$$\arg\min_{b\geq b_{\min}} V(0, b, 1) = \arg\min_{b\geq b_{\min}} \frac{1}{\exp(\frac{1}{b})-1} = \arg\max_{b\geq b_{\min}} \exp(\frac{1}{b}) = b_{\min} \tag{22}$$

$$\therefore v_{\min} = \frac{1}{\exp(\frac{1}{b_{\min}})-1}. \tag{23}$$

**Substitute the obtained $v_{\min}$ for getting the worst case** From the result of Equation 23 and Equation 13, we have the bound of $\epsilon$, which always guarantee $\epsilon$-pDP. Here, we assume the case the $i$-th player does his best to guarantee $\epsilon$-pDP and choose $b_i = \infty$. Then, we have

$$\frac{\min\left(m_{i,q(d_i)}\right) + \frac{|\mathcal{D}|-1}{\exp(1/b_{\min})-1}}{\frac{|\mathcal{D}|-1}{\exp(\frac{1}{b_{\min}})-1}} = \frac{1 + \frac{|\mathcal{D}|-1}{\exp(1/b_{\min})-1}}{\frac{|\mathcal{D}|-1}{\exp(1/b_{\min})-1}} \leq \exp(\epsilon) \tag{24}$$

$$\therefore \epsilon \geq \ln\left(\frac{1 + \frac{|\mathcal{D}|-1}{\exp(1/b_{\min})-1}}{\frac{|\mathcal{D}|-1}{\exp(1/b_{\min})-1}}\right), \tag{25}$$

which can be equivalently written by

$$b_{\min} \geq \frac{1}{\log\left(1 + (|\mathcal{D}|-1)(\exp(\epsilon)-1)\right)}. \tag{26}$$

In Equation 25, we have $m_{i,q(d_i)} \geq 1$, where the equality holds when $b_i = \infty$ and the PDF is uniform. Finally, there always exists at least one choice to improve the DP guarantee payoff for all elements.

**The strategy is improved to finally guarantee $\epsilon$-pDP for all elemnts** Before the strategy set satisfies the $\epsilon$-pDP for all elements, we have

$$\min_{b_i} \Delta P_{i,\text{E}} \geq 1 > \max_{b_i} \Delta P_{i,\text{U}}. \tag{27}$$

Equation 27 proves that there exists at least a choice to improve the $\epsilon$-pDP guarantee for an element, when the $\epsilon$ is bounded like Equation 25, and by the definition of $P_\text{U}$. Therefore, players choose a strategy to improve $\epsilon$-pDP until guaranteeing for all elements.

**The Nash equilibrium ensures $\epsilon$-pDP for all elements**  Assume that the Nash equilibrium point $\left(b_i^*, b_{-i}^*\right)$ does not satisfy the $\epsilon$-pDP for all elements,

$$\Pi_i(b_i^*, b_{-i}^*) \geq \Pi_i(b_i, b_{-i}^*) \Rightarrow |\mathcal{D}| > \Pi_i(b_i^*, b_{-i}^*) \geq \max \Pi_i(b_i, b_{-i}^*) = \max_{b_i} \left(P_{i,\mathrm{E}} + P_{i,\mathrm{U}}\right) \quad (28)$$

$$= \max_{b_i} \left(P_{i-1,\mathrm{E}} + P_{i-1,\mathrm{U}} + \Delta P_{i,\mathrm{E}} + \Delta P_{i,\mathrm{U}}\right) = P_{i-1,\mathrm{E}} + P_{i-1,\mathrm{U}} + \max_{b_i} \left(\Delta P_{i,\mathrm{E}} + \Delta P_{i,\mathrm{U}}\right) \quad (29)$$

$$\geq P_{i-1,\mathrm{E}} + P_{i-1,\mathrm{U}} + 1 \geq P_{i-2,\mathrm{E}} + P_{i-2,\mathrm{U}} + 2 \geq \ldots$$
$$\geq \min_{i,(b_i, b_{-i})} \left(P_{i,\mathrm{E}} + P_{i,\mathrm{U}}\right) + |\mathcal{D}| = |\mathcal{D}|, \quad (30)$$

where Equation 28 follows the definition of Nash equilibrium and the definition of NVO game's payoff. Because the result in Equations 27 to 30 contradicts $(|\mathcal{D}| > |\mathcal{D}|)$, we show that the assumption in this paragraph is false. That is, the Nash equilibrium point $\left(b_i^*, b_{-i}^*\right)$ must satisfy the $\epsilon$-pDP for all elements.

# B  Approximated enumeration for NVO game via genetic algorithm

Carrying out a precise enumeration for the proposed game proved to be computationally daunting. In lieu of that, we adopted an approach grounded in evolutionary game theory. We conducted an AE algorithm by running simulations across numerous generations. This technique tracks the evolution of strategies over time, shedding light on promising strategies without the necessity of exhaustively probing every conceivable option.

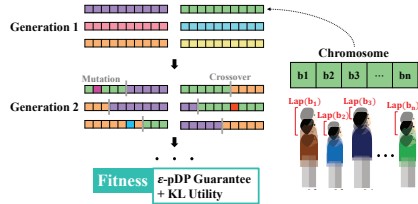

Figure 5: AE for the NVO game via genetic algorithm to find an NE point.

**Chromosome**  Chromosomes typically symbolize solutions to the specific optimization challenge being addressed. In the framework of the NVO games, each gene is representative of the variance variable $\mathbf{b}_i$ tied to the noise introduced to the query output for every sequential element.

**Fitness function**  Fitness function serves as the criterion for selecting the most suitable chromosomes that fulfill the specified criteria and can pass down their traits to offspring. Hence, we adopt the payoff $P(\mathcal{M}_\mathrm{I}, \mathcal{D})$ as our fitness function.

The initial generation's chromosomes are created by randomly selecting values within the variance space $\mathcal{V}$ for each gene. A larger population broadens the solution search space, minimizing the risk of local optima. Some high-fitness parents are retained in the offspring generation to avoid local optima. During offspring generation, random crossover points are used, and their optimal number can be determined via simulation. Mutation probability is set to balance between avoiding local optima and ensuring trait transfer. If the optimal fitness value stagnates across generations, it indicates a Nash equilibrium approximation. The current chromosome may be optimal, but due to randomness, other solutions might emerge.

With ample time, the AE algorithm has the potential to match the performance of the exact enumeration algorithm and attain an NE point. Our experiments continued for an extended period to ensure convergence. Nevertheless, there is no theoretical guarantee that an NE point is achievable within polynomial time.

# C  Experimental details

**Hyperparameters**  Our proposed BRD algorithm does not require specific hyperparameter settings. In the approximate enumeration via GA, we initially set the number of chromosomes in the population to 500, and for each generation, we involve 10 chromosomes in the mating process. We randomly designate 2 crossover points, and we introduce a 5 % probability for each gene to undergo

mutation. We employ a steady-state selection approach, retaining the top 5 parents with the highest fitness values for the next generation. We utilized PyGad (Gad, 2021) library for the implementation.

## C.1 REALISTIC VARIANCE VALUES

In Theorem 4.1, the value of $b_{\min}$ is sufficiently realistic as well as it is smaller than the noise added in the Laplace mechanism. For example, Apple is known to use $\epsilon$-DP with epsilon values ranging between 2 and 8 [5]. We assume that the tightest $\epsilon$ is given, i.e., $\epsilon = 2$, where the sensitivity after normalizing the query output is 1. The Laplace mechanism has an additive noise with the variance of $1/\epsilon = 0.5$. However, in Theorem 4.1, the value of $b_{\min}$ is 0.263 for a dataset of size 1,000. We kindly note that the variance is reduced as the dataset size increases, e.g., $b_{\min} = 0.208$ for a dataset of size 10,000.

## C.2 HARDWARE ENVIRONMENT

We conduct experiments using an AMD Ryzen Threadripper 1920X 12-Core Processor and 32 GB of RAM. Since there is no need for extensive parallel computations, GPU utilization is not required. To conserve computing resources and facilitate a fair comparison in terms of execution time on the same evaluation criteria, we exclusively relied on CPU computations.

## C.3 METRICS OF DATA STATISTICS

In this experiment, we use the following metrics related to data statistics:

- **KL divergence**: We measure the KL divergence between the probability distribution of the original dataset and the randomized dataset. The lower KL divergence indicates better preservation of the information of the original dataset.

- **L1 loss of standard deviation (SD)**: This metric measures the $\ell_1$ error between the standard deviation of the original dataset and the randomized dataset.

- **Jaccard index**: The Jaccard index is calculated by representing values in a probability distribution exceeding a certain threshold and then computing the intersection over union (IoU) of the two sets. This measure quantifies the similarity between two probability distributions, where a value closer to 0 indicates dissimilarity, while a value closer to 1 signifies similarity between the distributions. We set the threshold to 0.001 to examine the probability distribution of query output during experiments and select significant values.

- **Cosine similarity** (Furnas et al., 1988): The probability mass function can be viewed as a vector with probability values. We leverage the cosine similarity to measure the similarity between two probability distributions represented as vectors.

# D ADDITIONAL EXPERIMENTAL RESULTS

Here, we discuss additional results that were not included in the main manuscript. This section presents the results of randomized mechanisms applied to the height feature of the NBA player dataset.

## D.1 STATISTICAL UTILITY PRESERVATION

In Fig. 6, the random sampling query output of the original data, conventional Laplace mechanism, and NVO game (BRD and AE) is depicted. As similar to the result of the main manuscript, the NVO game better preserves the probability distribution than the conventional Laplace mechanism, by executing per-instance noise optimization. By observing the subfigures, we can observe that the probability distribution of the Laplace mechanism is similar to the original dataset; however, the proposed NVO game better preserves the shape than with only using $\epsilon = 0.1$ than the Laplace mechanism of $\epsilon = 8$.

---

[5]https://www.apple.com/privacy/docs/Differential_Privacy_Overview.pdf

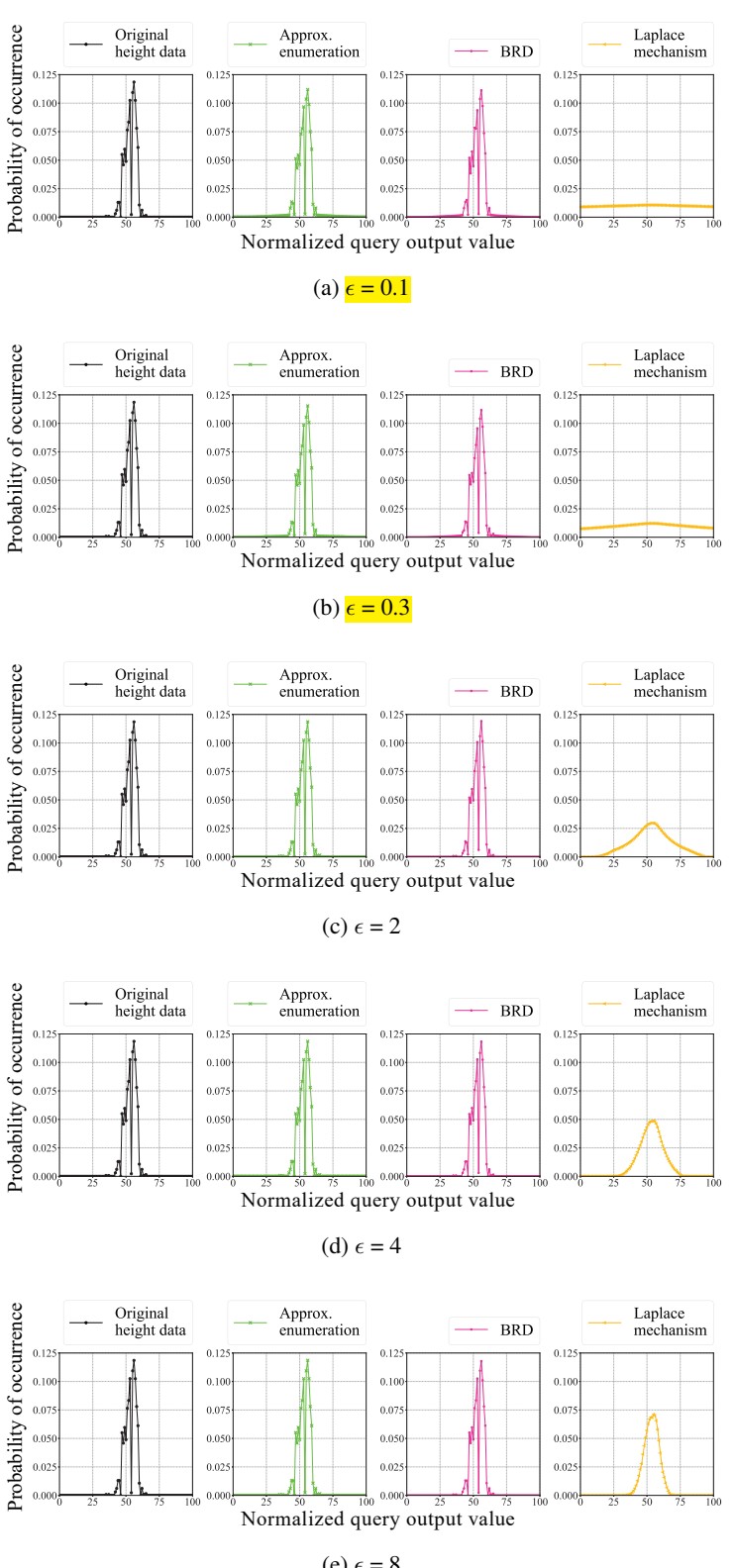

Figure 6: Comparison of query output probability distributions for the height data with each algorithm, when $\epsilon = 0.1, 0.3, 2, 4,$ and $8.$

Table 3: Each algorithm's average computation time, KL divergence, L1 loss of standard deviation, Jaccard index with a threshold 0.001, and cosine similarity are evaluated for the height data, for $\epsilon = 0.1, 0.3, 4, and 8.$ The modified query output distributions for all algorithms satisfy $\epsilon$-DP.

| Algorithm | Target $\epsilon$ | Comp. time ↓ (minutes) | KL divergence ↓ | L1 SD loss ↓ | Jaccard index ↑ (threshold=0.001) | Cos. similarity ↑ |
|---|---|---|---|---|---|---|
| BRD | $\epsilon$=0.1 | **8** | 0.0512 | 0.0126 | 0.4878 | 0.9989 |
| | $\epsilon$=0.3 | **5** | 0.0393 | 0.0122 | 0.5882 | 0.9994 |
| | $\epsilon$=4 | **5** | **0.0005** | 0.0016 | **1.0000** | 0.9999 |
| | $\epsilon$=8 | **5** | 0.0006 | 0.0017 | **1.0000** | 0.9999 |
| Approx. enum. | $\epsilon$=0.1 | 287 | **0.0475** | 0.0123 | 0.5000 | 0.9995 |
| | $\epsilon$=0.3 | 294 | **0.0248** | **0.0108** | **0.8261** | **0.9998** |
| | $\epsilon$=4 | 182 | 0.0006 | **0.0015** | **1.0000** | **0.9999** |
| | $\epsilon$=8 | 155 | **0.0001** | **0.0007** | **1.0000** | **0.9999** |
| Laplace mechanism (baseline) | $\epsilon$=0.1 | | 0.0475 | 0.0279 | 0.1980 | 0.3732 |
| | $\epsilon$=0.3 | | 0.0475 | 0.0279 | 0.1980 | 0.3732 |
| | $\epsilon$=4 | - | 0.5064 | 0.0216 | 0.4444 | 0.8299 |
| | $\epsilon$=8 | | 0.2401 | 0.0170 | 0.6451 | 0.9074 |

*Best: **bold**, second-best: underline.

In Table 3, the proposed NVO game and the Laplace mechanism are quantitatively evaluated in various statistical metrics. Similar to the results in the main manuscript, the proposed NVO game-based algorithms (BRD and AE) outperform the Laplace mechanism.

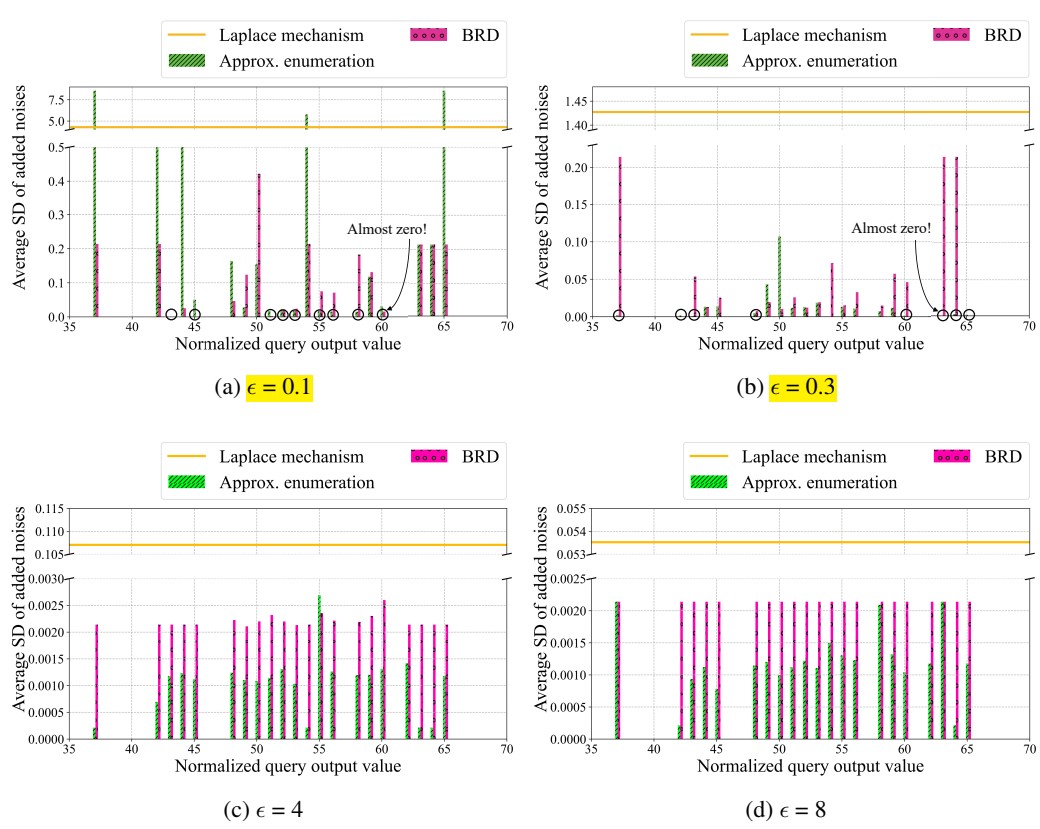

Figure 7: Distributions of average noise standard deviation for the income dataset for $\epsilon = 0.1, 0.3, 4,$ and 8.

In Fig. 7, we compare the average standard deviation of the added noise to each categorization bin of the conventional Laplace mechanism and the NVO-game-based algorithms. As depicted in the

figure, the NVO game adds lower variance at almost all bins, thereby having better data statistical utility.

## D.2 REGRESSION TASK

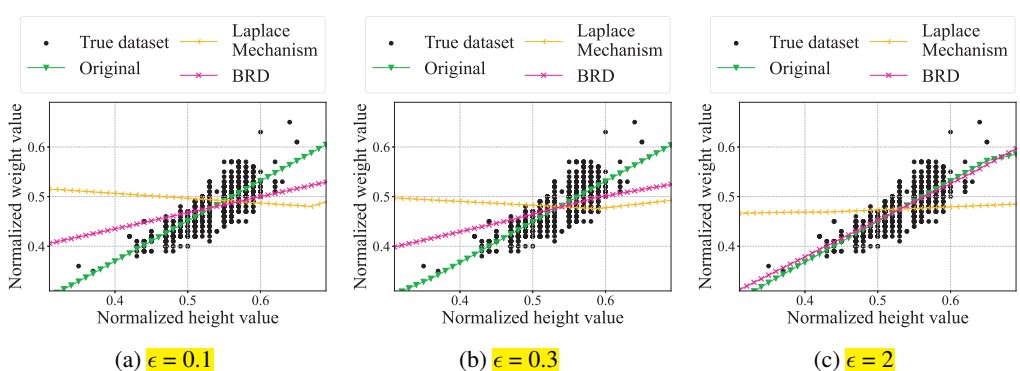

(a) $\epsilon = 0.1$      (b) $\epsilon = 0.3$      (c) $\epsilon = 2$

Figure 8: Linear regression result of 500 sampled data for each algorithm for $\epsilon = 0.1, 0.3$, and 2.

To evaluate the regression task of the proposed NVO game, we run the regression network with three layers. Similarly to the main manuscript, the network consists of ten parameters and ReLU activation functions. Figure 8 depicts the scatter diagram of the original dataset and the trained regression line, where the neural network input is height and the output is weight. For the value of $\epsilon = 0.1, 0.3$, and 2, the proposed NVO game closely preserves the regression line after applying the randomized algorithm (BRD).

## E EXPERIMENT 2: INCOME DATA

In addition to the NBA player dataset, we have conducted supplementary experiments on personal income data. Detailed experiment setups and the results are discussed in the remainder of this section.

### E.1 EXPERIMENT SETUP

**Personal income dataset** We utilize the test dataset of the personal income dataset, crafted by UC Irvine[6]. The number of data in the test dataset is 899. Similar to the NBA player dataset, the income values belong to one of the 101 categorization bins. We note that the single feature analysis is conducted here because there is no continuous feature in the dataset except income.

**Hyperparameters** Here, we use the same set of hyperparameters with the NBA player dataset's experiment.

### E.2 ANALYSIS

In Fig. 9, we can observe that the NVO-game-based algorithms (BRD and AE) better preserve the shape of the probability distribution of the income feature compared to the conventional Laplace mechanism.

The qualitative results can be found in Table 4. The approximate enumeration algorithm demonstrates an ability to maintain data statistics that were nearly equivalent, albeit at a computational cost approximately 100-150 times higher, requiring roughly 270 generations. The BRD algorithm achieves similar performance much more efficiently. The BRD algorithm achieved up to a 99.71 % improvement in KL utility than the Laplace mechanism, while guaranteeing 4-DP for every element, on the income dataset.

---

[6]https://www.kaggle.com/datasets/mastmustu/income

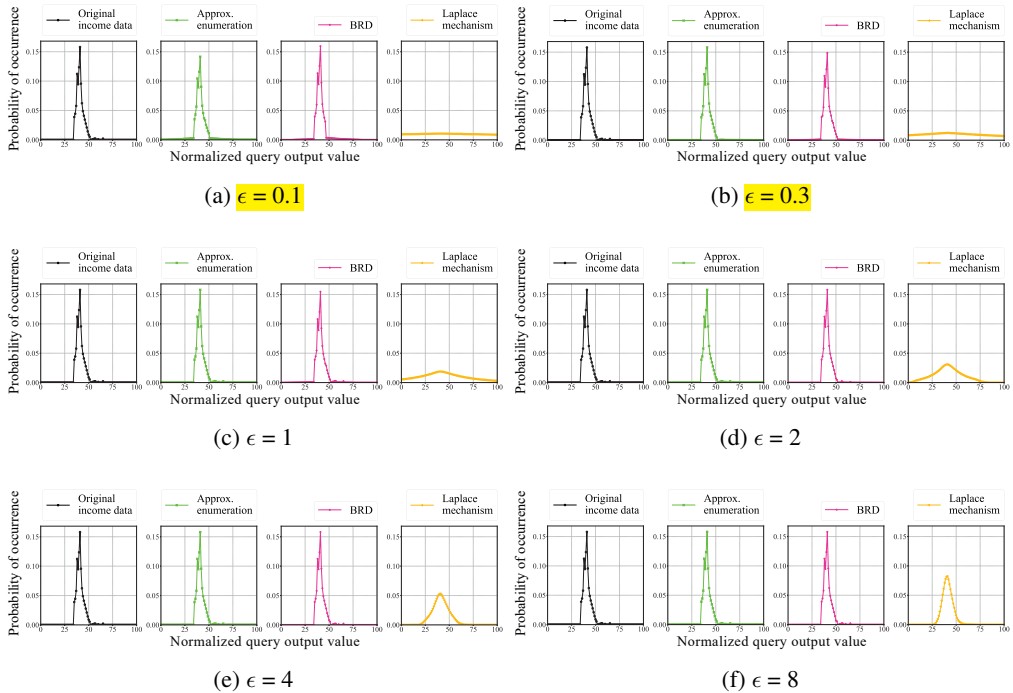

Figure 9: Comparison of query output probability distributions for the income data with each algorithm, when $\epsilon = 0.1, 0.3, 1, 2, 4,$ and $8$.

For the comparison of the per-instance noise variance, we depict the average standard deviation of the added noise in Fig. 10. Similar to the results in the NBA player dataset, the proposed NVO game allocate different amount of noise by considering its probability mass, thereby having better statistical utility.

From these additional results, we confirm that the proposed NVO game concisely outperforms the conventional Laplace mechanism.

## F  EXPERIMENT 3: LARGE INCOME DATA

Essentially, individual privacy is harder to guarantee with smaller datasets due to the increased significance of a data point. Hence, opting for smaller datasets makes privacy assurance more challenging. As datasets grow larger, ensuring privacy becomes comparatively easier. For these reasons, we conduct experiments on a dataset of approximately 1000 in size, but to demonstrate scalability, we also perform experiments on a dataset ten times larger, comprising 10,000 instances.

### F.1  EXPERIMENTAL SETUP

**Credit profile dataset**   We utilize the test dataset of the credit profile dataset [7]. We conduct our experiments with randomly sampled a cohort of ten thousand individuals. We note the correlation between age and income, and we exploit those two features in our experiments. Similar to the NBA player dataset, the income and age values belong to one of the 101 categorization bins.

**Hyperparameters**   Here, we use the same set of hyperparameters with the NBA player dataset's experiment.

---

[7]https://www.kaggle.com/datasets/yashkmd/credit-profile-two-wheeler-loan-dataset/

Table 4: Each algorithm's average computation time, KL divergence, L1 loss of standard deviation, Jaccard index with a threshold ~~0.015~~ ~~0.001~~, and cosine similarity are evaluated for the income data. The modified query output distributions for all algorithms satisfy $\epsilon$-DP.

| Algorithm | Target $\epsilon$ | Comp. time ↓ (minutes) | KL divergence ↓ | L1 SD loss ↓ | Jaccard index ↑ (threshold=0.001) | Cos. similarity ↑ |
|---|---|---|---|---|---|---|
| BRD | $\epsilon$=0.1 | **3** | 0.0697 | **0.0015** | **0.5758** | 0.9933 |
| | $\epsilon$=0.3 | **2** | 0.0376 | 0.0083 | **0.8500** | 0.9994 |
| | $\epsilon$=1 | **1** | 0.0223 | 0.0068 | **0.9091** | 0.9997 |
| | $\epsilon$=2 | **2** | 0.0087 | **0.0003** | 0.9000 | **0.9999** |
| | $\epsilon$=4 | **2** | 0.0014 | **0.0003** | **1.0000** | **0.9999** |
| | $\epsilon$=8 | **1** | 0.0013 | 0.0007 | **1.0000** | 0.9999 |
| Approx. enum. | $\epsilon$=0.1 | 183 | **0.0203** | 0.0110 | 0.5135 | **0.9989** |
| | $\epsilon$=0.3 | 235 | **0.0179** | **0.0012** | **0.8500** | **0.9999** |
| | $\epsilon$=1 | 148 | **0.0202** | **0.0061** | **0.9091** | **0.9998** |
| | $\epsilon$=2 | 204 | **0.0061** | 0.0026 | **1.0000** | **0.9999** |
| | $\epsilon$=4 | 178 | **0.0010** | 0.0017 | **1.0000** | **0.9999** |
| | $\epsilon$=8 | 164 | **0.0000** | **0.0000** | **1.0000** | **1.0000** |
| Laplace mechanism (baseline) | $\epsilon$=0.1 | | 1.9261 | 0.0279 | 0.1980 | 0.3589 |
| | $\epsilon$=0.3 | | 1.7909 | 0.0278 | 0.1980 | 0.4069 |
| | $\epsilon$=1 | | 1.3952 | 0.0274 | 0.1980 | 0.5529 |
| | $\epsilon$=2 | - | 0.9373 | 0.0261 | 0.2778 | 0.6959 |
| | $\epsilon$=4 | | 0.4758 | 0.0230 | 0.4419 | 0.8369 |
| | $\epsilon$=8 | | 0.1736 | 0.0178 | 0.5806 | 0.9378 |

*Best: **bold**, second-best: underline.

## F.2 STATISTICAL UTILITY PRESERVATION

In Fig. 11, the random sampling query output of the original data, conventional Laplace mechanism, and NVO game (BRD) is depicted. As similar to the result of the main manuscript, the NVO game better preserves the probability distribution than the conventional Laplace mechanism, by executing per-instance noise optimization. As the dataset size increased, the AE took an excessively long time to converge, preventing us from confirming its convergence within a reasonable timeframe. Consequently, we omitted its results from our findings.

Table 5: Each algorithm's average computation time, KL divergence, L1 loss of standard deviation, Jaccard index with a threshold 0.001, and cosine similarity are evaluated for the large income data, for $\epsilon = 1$ and 2. The modified query output distributions for all algorithms satisfy $\epsilon$-DP.

| Algorithm | Target $\epsilon$ | Comp. time ↓ (minutes) | KL divergence ↓ | L1 SD loss ↓ | Jaccard index ↑ (threshold=0.001) | Cos. similarity ↑ |
|---|---|---|---|---|---|---|
| BRD | $\epsilon$=1 | **331** | **0.0000** | **0.0000** | **1.0000** | **1.0000** |
| | $\epsilon$=2 | **633** | **0.0001** | **0.0004** | **1.0000** | **1.0000** |
| Laplace mechanism (baseline) | $\epsilon$=1 | - | 0.9490 | 0.0190 | 0.3069 | 0.6876 |
| | $\epsilon$=2 | | 0.5668 | 0.0173 | 0.3333 | 0.8224 |

*Best: **bold**.

In Table 5, the proposed NVO game and the Laplace mechanism are quantitatively evaluated in various statistical metrics. Similar to the results in the main manuscript, the proposed NVO game-based algorithm (BRD) outperforms the Laplace mechanism. Furthermore, for larger datasets, due to the lower individual instance contribution, privacy is better preserved, allowing us to ensure a more robust statistical utility while maintaining the same $\epsilon$-DP guarantee.

For extremely large datasets, our proposed method incurs high-order computational complexity for the $\epsilon$-pDP guarantee, scaling as $O(|\mathcal{D}|^2)$. To mitigate this, one approach could be to group data points with identical query outputs, allowing for computational reduction through the addition of uniform noise.

In Fig. 12, we compare the average standard deviation of the added noise to each categorization bin of the conventional Laplace mechanism and the NVO-game-based algorithm (BRD). As depicted

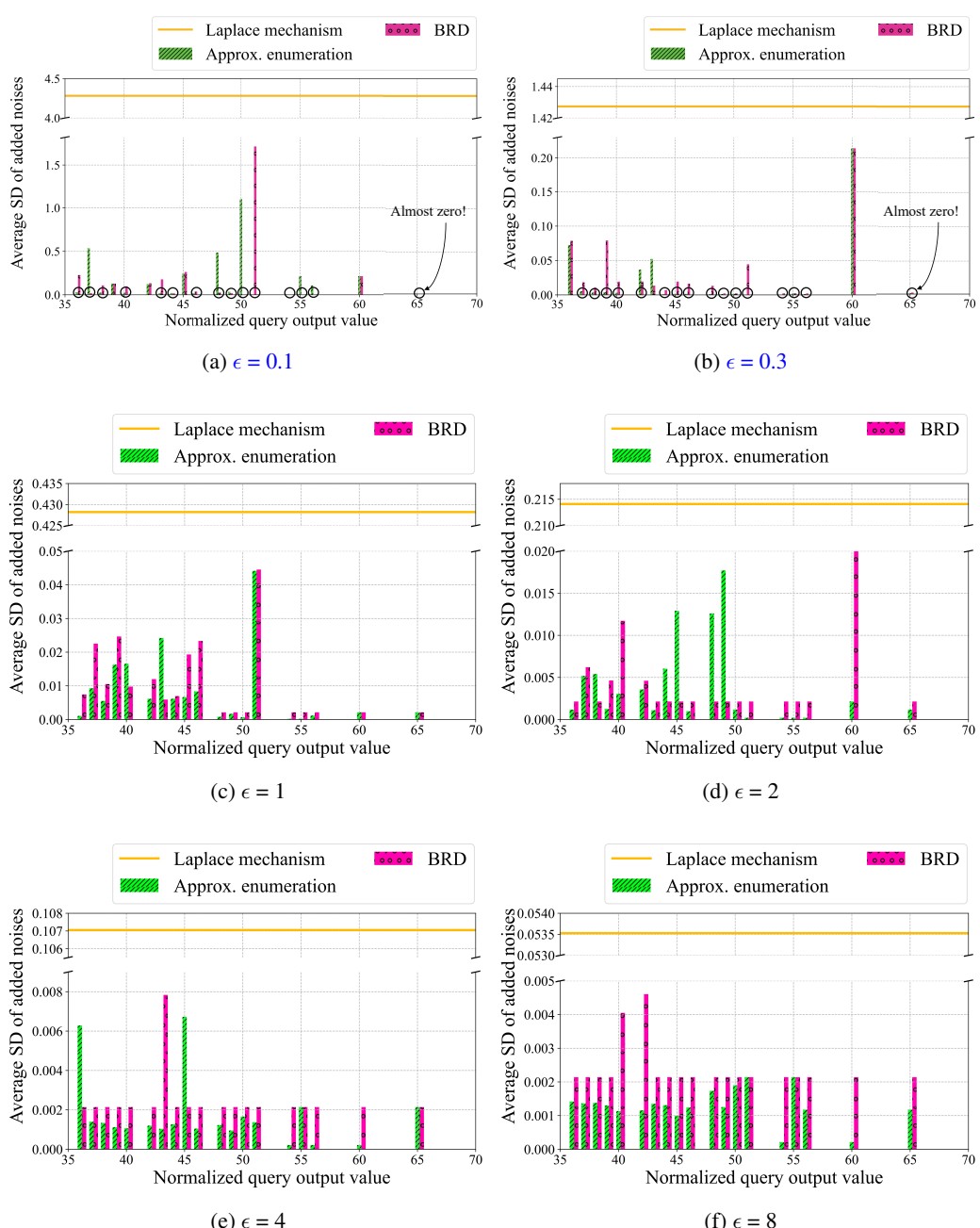

Figure 10: Distributions of average noise standard deviation for the height dataset for $\epsilon$ =0.1, 0.3, 1, 2, 4, and 8.

in the figure, the NVO game adds lower variance at all bins, thereby having better data statistical utility.

### F.3    REGRESSION TASK

To evaluate the regression task of the proposed NVO game, we run the regression network with three layers. Similarly to the main manuscript, the network consists of ten parameters and ReLU activation functions. Figure 13 depicts the scatter diagram of the original dataset and the trained

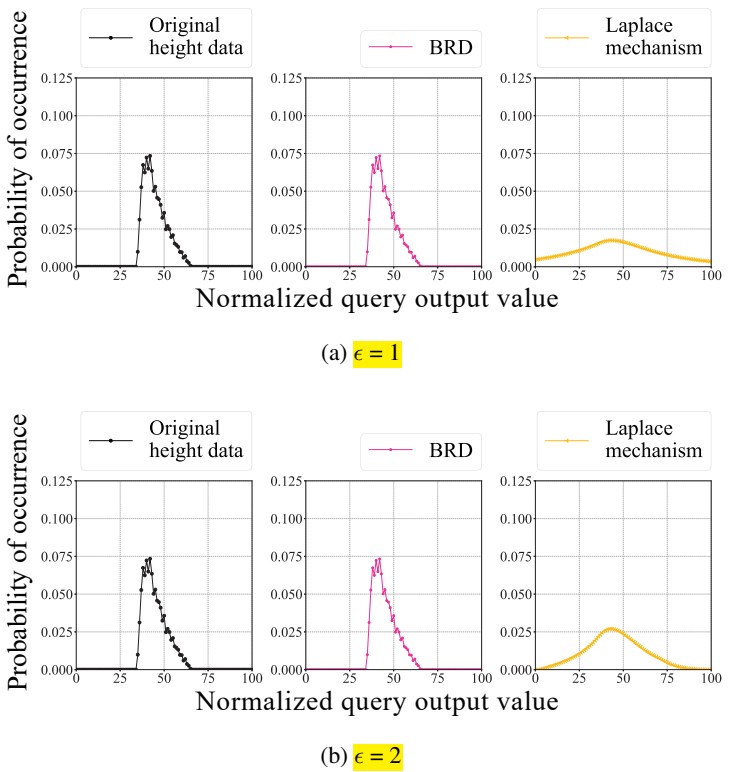

Figure 11: Comparison of query output probability distributions for the large income data with each algorithm, when $\epsilon = 1$ and 2.

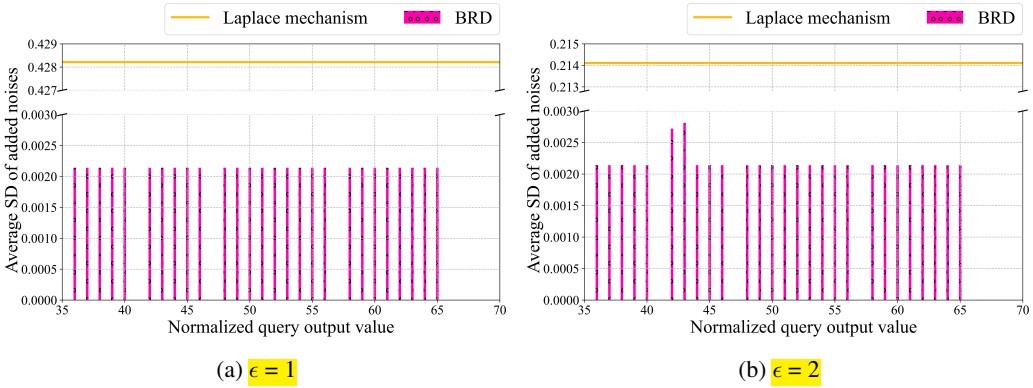

Figure 12: Distributions of average noise standard deviation for the large income dataset for $\epsilon = 1$ and 2.

regression line, where the neural network input is age and the output is income. For the value of $\epsilon=1$ and 2, the proposed NVO game closely preserves the regression line after applying the randomized algorithm (BRD). In regression tasks as well, we observe improved data characteristics for the same $\epsilon$-DP when dealing with larger dataset sizes.

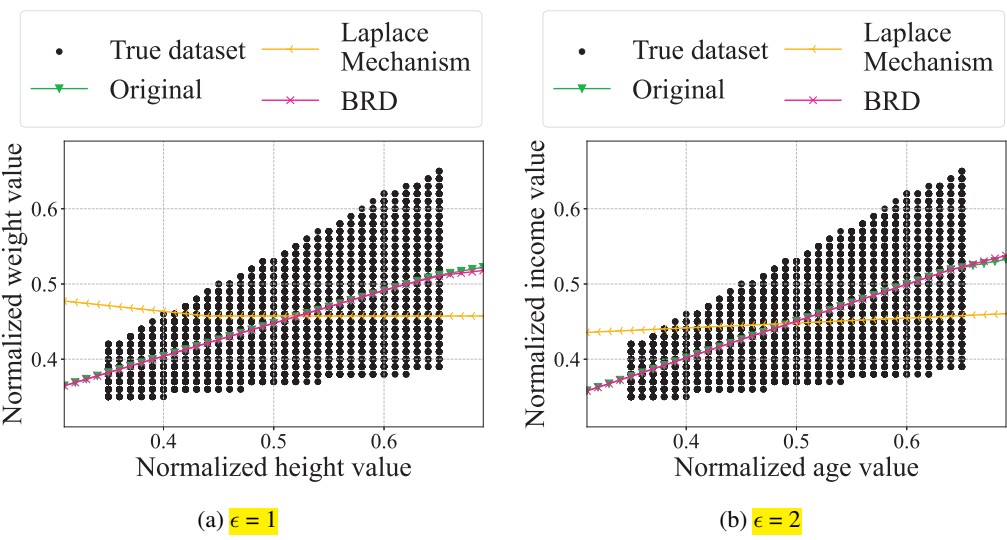

Figure 13: Linear regression result of 500 sampled data for each algorithm for $\epsilon = 1$ and 2.

