# OpenReview forum: "Differentially Pivate Per-Instance Additive Noise Mechanism: A Game Theoretic Approach"
_ICLR.cc/2024/Conference — Submitted to ICLR 2024_

### Official Review · Reviewer_HRkt · 2023-10-28

**Soundness:** 3 good
**Presentation:** 4 excellent
**Contribution:** 3 good
**Rating:** 6
**Confidence:** 4

**Summary:**

This paper presents a novel perspective to solve the pDP problem by a game-theoretical modeling. In particular, they introduce a per-instance noise variance optimization (NVO) game, which is designed to find suitable non-identical per-instance additive Laplace noises within a dataset. The authors use two algorithms to derive strategies for achieving the NE: 1) best response dynamics (BRD).1) best response dynamics (BRD), and 2) an approximate enumeration (AE) using a genetic algorithm. The paper demonstrates the efficacy of the NVO game on various statistical metrics and shows that it can achieve better statistical utility while maintaining the same level of DP as the conventional Laplace mechanism.

**Strengths:**

1. The problem of pDP is well motivated, considering that conventional DP often introduces substantial noise into the dataset, which can significantly diminish its statistical utility. The introduction of a game-theoretical modeling approach is indeed a novel and well-suited method for tackling this challenge, with the aim of optimizing per-instance noise.

2. The paper is commendably presented and effectively communicates its core contributions. It adeptly addresses fundamental questions within this domain, namely, 1) how to ensure the preservation of statistical utility, and 2) whether this alternative modeling approach maintains a sufficiently high level of privacy protection.

**Weaknesses:**

1. The experiments conducted in this study were carried out with a relatively low level of privacy protection, specifically $\epsilon = \{1,2,4,8\}$. It is worth noting that, in practical scenarios, an even lower value of $\epsilon$ (i.e., $\epsilon<1$) is often preferred. I would greatly appreciate it if the authors could either provide further justification for their choice of $\epsilon$ or consider conducting additional experiments with smaller values of $\epsilon$ to address this concern comprehensively.

2. I am genuinely curious about the computational cost associated with this approach. The datasets used in the experiments consisted of roughly one thousand samples, and the variance set was defined as relatively small. For the best response dynamics (BRD) approach, it would be valuable if the authors could present the computational time required. Regarding the approximate enumeration (AE) approach, I have some reservations about its privacy protection level, as Theorem 4.1 can only guarantee privacy protection for a Nash Equilibrium (NE) point. However, there is a possibility that AE may fail to obtain any NE point, which raises questions about the robustness of privacy protection in this context.

**Questions:**

See weakness.

---

> ### Author Response · Authors · 2023-11-21
> **Response to Reviewer HRkt**
>
> Thank you for your constructive feedback on additional numerical results and computational cost analysis. To cope with the comments, we added new experiments with smaller values of $\epsilon$. Also, as the reviewer suggested, we added computational cost analysis. We hope that the changes we made in the revised manuscript are satisfactory.
>
> ---
>
> **1. Additional experiments with smaller $\epsilon$.**
>
> > Comment: The experiments conducted in this study were carried out with a relatively low level of privacy protection, specifically $\epsilon$ =1, 2, 4, 8. It is worth noting that, in practical scenarios, an even lower value of $\epsilon$ (i.e., $\epsilon$<1) is often preferred. I would greatly appreciate it if the authors could either provide further justification for their choice of e or consider conducting additional experiments with smaller values of e to address this concern comprehensively.
>
> Answer: We sincerely appreciate this suggestion. The authors kindly note that $\epsilon>1$ is widely used for practical DP use cases. For example, [Apple](https://www.apple.com/privacy/docs/Differential_Privacy_Overview.pdf) is known to use $\epsilon$-DP with epsilon values ranging between 2 and 8.
>
> However, some previous papers have provided results for $\epsilon<1$ in extreme cases. Therefore, we conducted additional experiments for the cases where $\epsilon=0.1$ and $\epsilon=0.3$.
>
> - For additional experiments, please refer to Figures 6, 7, 8, 9, and 10. Based on the newly added results, we found that the proposed NVO game still better preserves utility while guaranteeing $\epsilon$-pDP. In these cases, the Laplace mechanism provides a uniform distribution, with almost zero utility. We added the results in Appendices D and E, because of the page limit.
>
> ---
>
> **2. Computational cost analysis**
> > Comment: I am genuinely curious about the computational cost associated with this approach. The datasets used in the experiments consisted of roughly one thousand samples, and the variance set was defined as relatively small. For the best response dynamics (BRD) approach, it would be valuable if the authors could present the computational time required. Regarding the approximate enumeration (AE) approach, I have some reservations about its privacy protection level, as Theorem 4.1 can only guarantee privacy protection for a Nash Equilibrium (NE) point. However, there is a possibility that AE may fail to obtain any NE point, which raises questions about the robustness of privacy protection in this context.
>
> Answer: Thank you for providing new suggestions. The reviewer raised questions about computational complexity of BRD approach and robustness of privacy preserving in AE approach. The bellows are one-by-one responses to the suggestions.
>
> **Computational complexity of BRD approach**: First of all, we kindly remind you that the wall-clock time of the BRD algorithm is available in Tables 1, 3, 4, and 5 for various datasets, including a newly added large dataset (credit profile dataset). Moreover, for computational complexity analysis, we first bring a fact from proof of Theorem 4.1 that in each round of the game, at least one player has a change of incrementally guaranteeing the pDP of a data instance. Thus, for privacy preserving, the maximum number of required steps is $|\mathcal{D}|$. Since the computational complexity of the payoff computation in each step of a round is $O(|\mathcal{D}|)$, the worst computational complexity of the BRD algorithm is $O(|\mathcal{D}|^2)$.
>
> For extremely large datasets, the required computation time becomes a limitation of our work. In such scenarios, it might be necessary to consider methods such as grouping data points with identical query output values to reduce computation by adding the same noises.
> Yet, additional future work is required to develop a better solution.
>
> To comply with the comment, we have revised the manuscript.
> - Added statements in **Limitation** paragraph in Section 7.
> - Add a paragraph **Convergence of BRD toward NE** in Section 5.1
>
> **Failure of AE to obtain NE point**:
> With ample time, the AE algorithm has the capability to reach a level of performance comparable to that of the exact enumeration algorithm, achieving an NE point. We extended our experiments for an extended duration to ensure convergence. However, there's no theoretical assurance of attaining an NE point within a polynomial time frame. In our revised manuscript, we relocated the detailed explanations of the AE algorithm to Appendix B. It's worth noting that we included this algorithm to assess the effectiveness of a non-greedy approach within our proposed NVO game when provided with ample computational resources.
>
> We strongly agree that we should address the exact solution that satisfies pDP. Thus, we additionally added more discussions of convergence in the BRD algorithm.
>
> - To reflect the comment, we moved the AE algorithm part into Appendix B on page 14, since AE is no longer our main focus.

---

> > ### Comment · Reviewer_HRkt · 2023-11-22
> >
> > The answers address the reviewer's concerns and the reviewer would like to keep the positive scores for this work.

---

> > > ### Author Response · Authors · 2023-11-22
> > > **Response to Reviewer HRkt**
> > >
> > > We sincerely thank the reviewer for his/her prompt response. We are happy that the reviewer's concerns have been clearly resolved and that they have maintained a positive attitude.
> > >
> > > It may be difficult, but all we have left to do is not lose hope and wait for a decision.
> > >
> > > We appreciate **the time and effort you devoted to reviewing our manuscript**, regardless of the final decision on its acceptance.
> > >
> > > Thank you!

---

### Official Review · Reviewer_UXaT · 2023-10-30

**Soundness:** 2 fair
**Presentation:** 2 fair
**Contribution:** 2 fair
**Rating:** 3
**Confidence:** 4

**Summary:**

This paper investigates the problem of computing good noise variances for a Laplace mechanism used to implement per-instance pure DP.  There is a well-known trade-off between the amount of noise and the level of privacy achieved, and so generally, the goal is to minimize the amount of noise required to achieve a certain privacy level. In the regular pure DP case in full generality, the amount of noise to be added is easily computed, but the per-instance definition leads to significant complications.

The authors frame this as a game and implement a best-response dynamics to compute a good noise variance. Results are compared in simulations to some baselines

**Strengths:**

* The algorithm seems correct and intuitive
* The performance does seem to be good when compared to baselines
* The approximate enumeration baseline is as interesting as the main result

**Weaknesses:**

This paper certainly has its redeeming qualities, but the negatives cannot be ignored. I found this paper very difficult to read due to the poor writing and errors.

I think the game theoretic framing is not meaningful. Perhaps it was included as a way to justify the optimization procedure. I would prefer if it was removed because I believe it does not add anything to the paper, but makes things more confusing.

My interpretation of what you are doing is setting two objective functions for both the utility and privacy, then you are optimizing their sum in a greedy way. With this understanding, I think this work falls below the standard for this venue.

There are some errors, even in very important mathematical statements, for example, as written, the LHS of (1) is always 0 since $z \in \mathcal{Z}$. I had to read the original reference to understand what pDP was. Please address this.

**Questions:**

1.  Why did you decide to use a game-theoretic presentation?

2. What does (1) mean, as the current statement seem meaningless?

3.  What does it mean to take a union of vectors as in Algorithm 1?

4. What is $\mathbf{b}_j$ in the inner for loop of Algorithm 1?

---

> ### Author Response · Authors · 2023-11-21
> **Response to Reviewer UXaT (Part 1)**
>
> Thank you for your valuable feedback on our manuscript. In light of your comments, we have added an explanation about our choice to use game-theoretic interpretations. While there are various approaches to introduce our work, such as optimization theory, we opted for the method we believed would be most accessible and familiar to our audience. We acknowledge that our initial explanation, particularly regarding Figure 1, had certain shortcomings.
> Also, we change the vector-form data representation $\mathbf{d}_i$ into a scalar representation $d_i$ for simplicity. Accordingly, we have revised the manuscript for improved clarity and a more effective presentation of our approach. We earnestly hope that these modifications will meet the reviewer's expectations and enhance the overall quality of our manuscript. In addition, we revised the fatal errors you pointed out.
>
> ---
>
> **1. Use of game theory**
> > Comment: Why did you decide to use a game-theoretic presentation?
>
> We are sincerely sorry for the complicated interpretation.
> We use game-theoretic presentation because we think it is familiar to the audience.
> As a result, the proposed algorithm is a greedy algorithm for updating each data instance’s additive noise.
> Here, we kindly note that we mainly contributed by showing the NE point of the NVO game guarantees the $\epsilon$-pDP for all data instances, where the greedy algorithm cannot guarantee $\epsilon$-pDP if Equation 9 does not hold.
>
> Also, we note that the proposed work has the following challenges in optimization perspective.
>
> We can represent the target optimization as a constraint problem.
>
> $$\min U(\mathcal{D}, \mathcal{M}(\mathcal{D})) ~~~ \textnormal{s.t.} \left|\ln\frac{\Pr[\mathcal{M}(\mathcal{D}) \in S]}{\Pr[\mathcal{M}\textcolor{blue}{(\mathcal{D} \setminus \{\mathbf{d}\})} \in S]}\right| \leq \epsilon, \forall \mathbf{d} \in \mathcal{D}, \forall S\in\mathnormal{Range}(\mathcal{M})$$
>
> - Initially, the continuous nature of the domain $S$ poses a unique challenge, differing from that in $\epsilon$-DP. In this context, it's necessary to confirm the $\epsilon$-pDP condition for each instance and each point within $S$, an endeavor impractical to accomplish in polynomial time. To address this issue, we have employed data categorization in our approach.
> - Furthermore, the variable $\mathbf{b}_i$ exhibits both continuity and a complex interdependence. While gradient descent is a common technique for finding local optima in standard optimization problems, our situation is complicated by the constraint being a non-differentiable function, which makes typical projection methods ineffective. As a solution, we have opted to moderate the variance of $b$, thereby simplifying the problem.
>
> Regrettably, despite the desire to pivot the problem definition from game theory to optimization theory, substantial portions had already been developed centered on game theory. Fully acknowledging the difficulty in comprehending game theory-centric explanations, we extensively revised Section 4 of the paper, although every change cannot be highlighted. We made substantial modifications to enhance the clarity of explanations. We earnestly hope that these revisions bring greater satisfaction to your understanding.
>
> We sincerely apologize for confusion, and we hope these revisions significantly enhance the paper's comprehensibility.
>
> ---
>
> **2. Correction of Equation 1**
> > Comment: What does (1) mean, as the current statement seems meaningless?
>
> We deeply apologize for an oversight in a crucial definition that permeates the entire paper. We sincerely appreciate your interest in our work.
> We have rectified a fatal error in Definition 3.1 as follows.
>
> Definition 3.1 ($\epsilon$-pDP)
> A randomized mechanism, denoted by $\mathcal{M}$, has a range $\mathcal{R}(\mathcal{M})$. For a fixed dataset $\mathcal{Z}$ and a fixed data instance $z\in\mathcal{Z}$, the mechanism $\mathcal{M}$ meets $\epsilon$-pDP, if the following condition holds:
>
> $$
> \bigg|\ln\frac{\Pr[\mathcal{M}(\mathcal{Z}) \in S]}{\Pr[\textcolor{blue}{\mathcal{M}(\mathcal{Z} \setminus \{z\})} \in S]}\bigg| \leq \epsilon, ~ \forall S \subseteq \textnormal{Range}(\mathcal{M}).
> $$
>
>
> We have uniformly rectified other sections to maintain consistency.
>
> **Remark:** The variance in mathematical notations compared to Wang (2019) lies in Definition 3.1. Consequently, it deals with the difference in probability of query outputs based on the presence or absence of a single data point, thus remaining equivalent meaning.

---

> ### Author Response · Authors · 2023-11-21
> **Response to Reviewer UXaT (Part 2)**
>
> **3. Correction of Algorithm 1**
> > Comment: What does it mean to take a union of vectors as in Algorithm 1?
> > Comment: What is bj in the inner for loop of Algorithm 1?
>
> We sincerely appreciate your corrections. The authors agree that the notation is really confusing. We deeply apologize for this typo in the explanation of the main algorithm. We corrected it to make the notations clearer.
>
> **Union of vectors in Algorithm 1**:
> Thank you for this question. We have a mistake in this notation, we forgot to exploit the set notation. We replace $\mathcal{D}=\bigcup_{i=1}^{|\mathcal{D}|}d_{i}$ by $\mathcal{D}=\set{d_{i}|i=1,...,m}$
>
> and $\mathcal{V} =\bigcup_{i=1}^{|\mathcal{V}|}{v_{i}}$ into $\mathcal{V}=\set{v_{i}|i=1,...,n}$.
>
>
> **$b_j$ in the inner for loop of Algorithm 1**: We apologize sincerely for the error in depicting $b$ (now $v$) as a vector when it should have been a scalar. To clarify, $b_j$ (now $v_j$) refers to the variance value corresponding to the $j$-th element in the variance set $\mathcal{V} = \set{v_{1}, v_{2}, …, v_{n}}$.

---

### Official Review · Reviewer_xkkw · 2023-11-01

**Soundness:** 2 fair
**Presentation:** 1 poor
**Contribution:** 2 fair
**Rating:** 3
**Confidence:** 2

**Summary:**

This paper proposes an algorithm for optimizing the additive noise variances for achieving per-instance differential privacy, and then demonstrates the algorithm's performance on real data sets.

**Strengths:**

* The game-theoretic formulation of noise distribution optimization for DP (and variants of DP) is a new idea, to the best of my knowledge.
* There is an extensive set of experiments assessing the performance of the proposed algorithm and comparing with existing noise addition algorithms.

**Weaknesses:**

* Lack of problem formulation. A central premise of the paper is that "ensuring pDP for a particular data instance is inherently dependent on the noise distribution of other instances". However the claim is not formulated in mathematical terms, making the paper difficult to follow for those readers who are not already convinced of this claim before reading the paper. The lack of problem formulation also makes it confusing why existing noise mechanisms (for example the per instance Gaussian mechanism in Wang (2019)) are not desirable.

* Ambiguous scope. The first 2.5 pages of the paper give an impression that the new noise addition mechanism, similar to the Laplace mechanism for DP, is suitable for general queries, but on page 3 it is then stated that "we focus on the random sampling query" and $q$ is defined to be the random sampling query. Later, in Remark 4.1 on page 4, the utility function uses $q$ to refer to generic queries. It is ambiguous whether the paper's results are applicable to queries other than the random sampling query.

* Some minor issues:
     * In "Relation to ($\epsilon, \delta$)-DP" under Section 2, the citation for Gaussian mechanism is incorrect: Dwork and Roth (2014) is an expository work; the Gaussian mechanism appeared in the literature much earlier.
     * The definition of pDP is imprecise. Compared to the original definition in Wang (2019), Definition 3.1 in this paper does not explicitly fix the data set $\mathcal Z$
    * There is an incorrect statement above Definition 3.2: pDP holding for "every $z$ within $\mathcal Z$" does not guarantee DP, if the data set $\mathcal Z$ is fixed.

**Questions:**

* Can you formulate the difficulty of optimizing noise distribution for pDP in mathematical terms? What target is being optimized? What is the relation of your work to, for example, the Gaussian mechanism in Wang (2019)?

* Is your work, in the present form, limited to the random sampling query?

* Is the algorithm in Section 5.2 trying to find an approximate instead of an exact solution? If so, does the pDP guarantee hold for the approximate solution? It appears that we only know from Theorem 4.1 that the exact solution satisfies pDP.

---

> ### Author Response · Authors · 2023-11-21
> **Response to Reviewer xkkw (Part 1)**
>
> We greatly appreciate the constructive feedback on our manuscript. In response, we have first introduced an optimization problem and explored the challenges associated with solving it. Additionally, we've expanded our analysis to include the applicability of our approach to general statistical queries. Regarding the approximated enumeration (AE) method, we acknowledge that, unlike the Best Response Dynamics (BRD) algorithm, AE cannot mathematically guarantee an exact Nash Equilibrium (NE) within a finite time frame. To address this, we have shifted our focus in the main manuscript to emphasize the BRD algorithm over AE. The detailed discussion on AE has been moved to Appendix B. We sincerely hope that the reviewer will be satisfied with the changes we made in the revised manuscript.
>
> ---
>
> **1. Problem formulation should be addressed.**
> > Comment: Can you formulate the difficulty of optimizing noise distribution for pDP in mathematical terms? What target is being optimized? What is the relation of your work to, for example, the Gaussian mechanism in Wang (2019)?
>
> Answer: Thank you for this suggestion. To comply with the comment, we first added the following optimization problem in the main paper.
>
> $$\min U(\mathcal{D}, \mathcal{M}(\mathcal{D})) ~~~ \textnormal{s.t.} \left|\ln\frac{\Pr[\mathcal{M}(\mathcal{D}) \in S]}{\Pr[\mathcal{M}(\mathcal{D} \setminus \{d\}) \in S]}\right| \leq \epsilon, \forall d \in \mathcal{D}, \forall S\in\mathrm{Range}(\mathcal{M})$$
>
> where the objective function minimized represents the distributional difference between $q(\mathcal{D})$ and $q(\mathcal{M}(\mathcal{D}))$, with various metrics available.
>
> Our primary focus is on addressing the constraint of the problem, namely $\epsilon$-pDP. Consider the term $\left|\ln\frac{\Pr[\mathcal{M}(\mathcal{D}) \in S]}{\Pr[\mathcal{M}(\mathcal{D} \setminus \{d\}) \in S]}\right|$.
>
> - First, the domain of $S$ is continuous, which presents a challenge distinct from that in $\epsilon$-DP. Here, we must verify the $\epsilon$-pDP condition for each instance and each point within $S$, a task that is not feasible within polynomial time. To overcome this, our method incorporates data categorization.
> - Second, the variance parameter $v$ is continuous and intricately interlinked. In typical optimization problems, gradient descent is utilized to identify a locally optimal solution. However, in our case, the constraint represents a non-differentiable function, rendering common projection methods unsuitable. Therefore, we relax the diversity of $v$ to simplify and solve the problem effectively.
>
> We propose an NVO game, designed based on a relaxed version of our initial problem. Despite this relaxation, ensuring $\epsilon$-pDP remains a significant challenge. In Theorem 4.1, we demonstrate that our proposed NVO game successfully achieves $\epsilon$-pDP, provided that the condition (9) is met.
>
> **Relation to $\epsilon$-pDP study :**
> Wang's (2019) study drew attention to the varying levels of privacy protection in data instances when the same noise is added to each query output. However, it primarily focused on identifying and analyzing these issues, **without offering concrete solutions**. In contrast, our paper represents a pioneering effort in this field. We **formulate and address a problem that involves optimizing the addition of distinct noises to individual data instances**. This approach ensures a uniform level of pDP throughout the dataset while maintaining its statistical utility. We appreciate your feedback highlighting this critical aspect of our work. Accordingly, we have revised the paper to more prominently feature this significant contribution."
>
> - We have revised statements to reflect this discussion. (please refer to Section 1 on page 1, and Section 4.1 on page 4)

---

> ### Author Response · Authors · 2023-11-21
> **Response to Reviewer xkkw (Part 2)**
>
> **2. Extensibility to other queries.**
> > Comment: Is your work, in the present form, limited to the random sampling query?
>
> Answer: **It is definitely possible. The random sampling query is a fundamental query that encompasses all possible statistical queries.** More specifically, the random sampling query can capture the statistical distribution of a dataset. Therefore, based on the post-processing theorem, our proposed NVO game can answer all statistical queries that rely on this distribution.
>
> **Example:** Let's consider the NBA player height dataset. Suppose the query asks for the proportion of players with a height of 203 cm or above. In this scenario, we can derive the response from the distribution of the modified dataset through random sampling queries. In the original height dataset, the proportion of players with a height of 203 cm or above is 49.16%. If one player who is 210 cm tall is removed from the dataset, the proportion changes to 49.08%, with the difference being only 0.08%. However, after applying our algorithm, the calculated proportions are 48.84% with the player and 48.80% without the player.
>
> - To comply with the comment, we have added Remark 3.1 on page 3, and a new discussion in Section 7 on page 9 (paragraph **Extensibility to other queries**).
>
> ---
>
> **3. Guarantees of AE algorithms for the NE.**
>
> > Comment: Is the algorithm in Section 5.2 trying to find an approximate instead of an exact solution? If so, does the pDP guarantee hold for the approximate solution? It appears that we only know from Theorem 4.1 that the exact solution satisfies pDP.
>
> **AE Algorithm**: With **ample time**, the AE algorithm has the potential to match the performance of the exact enumeration algorithm and attain an NE point. Our experiments continued for an extended period to ensure convergence. Nevertheless, there is **no theoretical guarantee** that an **NE point is achievable within polynomial time**. In the updated version of our manuscript, we have relocated the specifics of **the AE algorithm to Appendix B**. It is important to mention that this algorithm was included to evaluate the performance of a non-greedy approach in our proposed NVO game when given ample computational time.
>
> We strongly agree that we should address the exact solution that satisfies pDP. Thus, we additionally added more discussions of convergence in the BRD algorithm.
>
> - To reflect the comment, we moved the AE algorithm part into Appendix B on page 14 and added this discussion.
>
> ---
>
> **4. Minor issues**
> > Comment: In "Relation to (e, delta)-DP" under Section 2, the citation for Gaussian mechanism is incorrect: Dwork and Roth (2014) is an expository work; the Gaussian mechanism appeared in the literature much earlier.
> Answer: We have revised the sentence to comply with the comment.
> - Page 2: The pioneer of DP, Dwork & Roth (2014), defined (ϵ, δ)-DP, demonstrating that the Gaussian mechanism can achieve this definition.
>
>
> > Comment: The definition of pDP is imprecise. Compared to the original definition in Wang (2019), Definition 3.1 in this paper does not explicitly fix the data set Z.
>
> Answer: Thank you for your careful comment. We added **fixed** dataset $\mathcal{Z}$ in the Definition 3.1.
>
> - Please refer to Definition 3.1 to see the changes.
>
> > Comment: There is an incorrect statement above Definition 3.2: pDP holding for "every $z$ within $\mathcal{Z}$" does not guarantee DP, if the data set $\mathcal{Z}$ is fixed.
>
> Answer: Thank you for this clarification again. To correct the statement, we have added a new condition “for all dataset $\mathcal{Z}$”.
> - Please refer to statements below Equation 1 and Equation 9.

---

### Official Review · Reviewer_eCZ8 · 2023-11-03

**Soundness:** 4 excellent
**Presentation:** 3 good
**Contribution:** 3 good
**Rating:** 6
**Confidence:** 2

**Summary:**

This paper proposed a game theoretical approach to achieve data dependent privacy guarantees. Specifically, based on the per-instance differential privacy (pDP) definition, a per-instance noise variance optimization (NVO) game is designed and the Nash equilibrium (NE) guarantees DP. An approximate enumeration (AE) algorithm or a best response dynamics (BRD) algorithm can be used to solve the Nash equilibrium.

=======after rebuttal======

I thank the authors for the response. I would like to maintain the borderline positive evaluation.

I think it is a cool idea to use the game theoretical approach for  instance DP. I cannot strongly champion this paper as I do not consider myself as an expert in either game theory or instance DP. Though the draft is improved during the rebuttal, the various initial sloppiness in instance DP and algorithmic convergence makes me less confident in raising the score and championing it.

**Strengths:**

As far as I know, the game theoretical approach for per-instance DP is new. The proposed approach looks technically solid. The privacy utility trade-off of pDP is better than the baseline Laplace \epsilon-DP.

**Weaknesses:**

Unfortunately, I am not an expert on either game theory or per-instance DP, so I would rather use this opportunity to ask questions below.

In general, my questions are around intuition, experiments and baseline methods.

**Questions:**

Could the authors provide more intuition of theorem 4.1: e.g., why does it hold intuitively; how realistic is condition (8)?

Could the authors comment on the guarantees of AE and BRD Algorithms in Section 5 for the NE? Please provide necessary intuition, or properly cite references if it is well known.

In Section 6 Experiment, it would be nice to actually verify the per-instance \epsilon is achieved using the game theoretical algorithms; the dataset does not seem to be large, and I would hope to see some discussion on the scalability of the approach.

Finally, I am a little surprised that the only baseline is the worst-case \epsilon-DP with Laplace noise. Are there no other pDP or data dependent DP methods to compare with?

---

> ### Author Response · Authors · 2023-11-21
> **Response to Reviewer eCZ8 (Part 1)**
>
> We sincerely appreciate your constructive feedback on our manuscript. To follow the comments, we mainly revised the manuscript by newly adding 1) intuition of Theorem 4.1, 2) guarantee of convergence to NE, and 3) experimental results for a large dataset. Also, we re-checked existing studies and found that there has been no data-instance-wise pDP mechanism. The point-by-point responses are given below. We believe that the revised manuscript can be more insightful to future readers.
>
>
> Thank you sincerely.
>
> ---
>
> **1. Intuition of Theorem 4.1.**
> > Comment: Could the authors provide more intuition of theorem 4.1: e.g., why does it hold intuitively; how realistic is condition (8)?
>
> Answer: To follow your suggestion, we bring Equation 9 (Eq. 8 in the original manuscript) here:
>
> $$
> b_{\min}\ge\frac{1}{\log(1+(|\mathcal{D}|-1)(\exp(\epsilon)-1))}.
> $$
>
> **Intuition**: From the above equation, as the value of $|\mathcal{D}|$ increases, we require smaller variance noise for guaranteeing $\epsilon$-pDP. Intuitively, it is evident that the more data instances there are in the dataset, the less influence each individual data instance has. On the other hand, if $\epsilon$ reaches zero, the query output with and without a data instance should be statistically the same. In other words, adding noise that tends toward infinite variance becomes necessary, resembling a uniform distribution of noise.
>
> - This discussion is added in Remark 4.2 on page 6.
>
> **Realistic variance values**: In Theorem 4.1, the value of $b_{\min}$ is sufficiently realistic but also smaller than the noise added in the Laplace mechanism. For example, Apple is known to use $\epsilon$-DP with epsilon values ranging between 2 and 8 [(link).](https://www.apple.com/privacy/docs/Differential_Privacy_Overview.pdf)
> If the smallest $\epsilon$ is given, i.e., $\epsilon=2$, the Laplace mechanism has an additive noise with the variance parameter of $1/\epsilon=0.5$, where the sensitivity is one. However, in Theorem 4.1, the proposed NVO game has a minimum value of variance parameter $b_{\min}$ by 0.263 for a dataset of size of 1,000.
> We kindly note that the variance is reduced as the dataset size increases, e.g., $b_{\min}=0.208$ for a dataset size of 10,000.
>
> - This discussion is added in Appendix C.1 on page 15.
>
> ---
>
> **2. Guarantees of AE and BRD algorithms for the NE.**
> > Comment: Could the authors comment on the guarantees of AE and BRD Algorithms in Section 5 for the NE? Please provide necessary intuition, or properly cite references if it is well known.
>
> **BRD Algorithm**: The authors have referenced [R1] in the revised manuscript, indicating that the BRD algorithm converges to an NE point for games falling into specific categories: potential games, weakly acyclic games, aggregative games, and quasi-acyclic games. Given that the NVO game is categorized as a potential game, the BRD algorithm effectively reaches an NE point for this game.
>
> **Intuition of BRD Algorithm toward NE**: The authors kindly inform that the intuition is in **BRD Algorithm in potential game** in Section 5. We bring the corresponding statements here. “Intuitively, as players opt for their best responses, either sequentially or simultaneously, the potential function's value rises, eventually peaking. The strategy at this peak is the game's NE.”
> - The intuition of the BRD algorithm’s convergence has been relocated to the paragraph **Convergence of BRD toward NE** on page 6.
>
> [R1] Boucher, Vincent, Selecting Equilibria Using Best-Response Dynamics (September 1, 2017). CRREP Working Paper Series 2017-09, Available at SSRN: https://ssrn.com/abstract=3175335 or http://dx.doi.org/10.2139/ssrn.3175335
>
> **AE Algorithm**:  Given ample time, the AE algorithm can perform equivalently to the enumeration algorithm, thereby reaching an NE point. We ran experiments for a sufficient duration until convergence was achieved. However, it has not been theoretically proven that an NE point can be obtained in polynomial time. In the revised manuscript, we relocate the details of AE into Appendix B. We kindly note that we add this algorithm to know the performance of a non-greedy algorithm in the proposed NVO game with sufficient computation time.
>
> - We relocated the section of the AE algorithm (Section 5.2) in Appendix B.

---

> ### Author Response · Authors · 2023-11-21
> **Response to Reviewer eCZ8 (Part 2)**
>
> **3. Verification of $\epsilon$-pDP on a larger dataset for showing scalability**
> > Comment: In Section 6 Experiment, it would be nice to actually verify the per-instance ε is achieved using the game theoretical algorithms; the dataset does not seem to be large, and I would hope to see some discussion on the scalability of the approach.
>
> Answer: Thank you for this suggestion. We kindly remind you that we have shown that the proposed NVO game can achieve $\epsilon$-pDP for all datasets if condition (9) is satisfied. From the obtained condition, larger datasets require lower minimum variance to guarantee $\epsilon$-pDP, as the significance of each data instance is getting smaller.
>
> On the other hand, practically, we strongly agree with the reviewer that the experimental results lack the scalability of larger dataset. Thus, we added new experimental results in Appendix F. From the newly added experiments, we confirm that the proposed NVO game still works well and outperforms the baseline mechanisms.
>
> - We have added new experimental results in Appendix F on page 20. Please refer to newly added figures and table: Table 5, Figure 11, Figure 12, and Figure 13.
>
> New Discussion for Limitation: In addition to presenting new results, we also discuss the limitations of our work when applied to large datasets. For extremely large datasets, our proposed method incurs high-order computational complexity for the $\epsilon$-pDP guarantee, scaling as $O(|\mathcal{D}|^2)$. To mitigate this, one approach could be to group data points with identical query outputs, allowing for computational reduction through the addition of uniform noise.
>
> - We have reflected this discussion in Section 7 on page 9 (**Limitation** paragraph), and Appendix F.2 on page 22.
>
> ---
>
> **4. Additional baseline methods**
> > Comment: Finally, I am a little surprised that the only baseline is the worst-case ε-DP with Laplace noise. Are there no other pDP or data-dependent DP methods to compare with?
>
> Answer: Thank you for this suggestion. Upon thorough review of existing studies, we confirm that while numerous algorithms exist for ensuring DP, none have yet introduced per-instance noise or pDP-inspired mechanisms. It is important to highlight that our proposed NVO game represents the first initiative to address per-instance noise mechanisms grounded in the pDP concept.
>
> - We have revised Section 1 on page 1 to highlight our contribution related to this comment.

---

### Author Response · Authors · 2023-11-21
**General Comment by Authors**

General Comment

Dear reviewers and AC,

We are sincerely grateful for your valuable feedback, as well as your efforts and time reviewing our manuscript.

As reviewers highlighted, our work proposes a novel per-instance differential privacy mechanism based on $\epsilon$-pDP as well as preservation of statistical utility. The numerical results show that the proposed NVO game better preserves statistical utility compared to baseline approaches.

Generally, the reviewers provide constructive feedback on our manuscript. Whether the review scores are high or not, the comments provided by the reviewers are really helpful for enhancing the quality of our manuscript. In response to the comments, we have carefully revised the manuscript as follows:

- **Guarantee of AE converging to NE**: Reviewers eCZ8, xkkw, and HRkt commonly addressed this comment. The genetic algorithm is a well-known algorithm that converges to a globally optimal solution in infinite time, theoretically. However, in the privacy preservation perspective, exact convergence is the most important factor. Thus, our opinion is to move the AE approach to Appendix B. In the numerical results, to compare with the greedy/non-greedy algorithms, we executed AE until the convergence with excessive generations and populations.

- **Difficulty of the problem**: Reviewers xkkw and UXaT commonly pointed out lack of explanations on the difficulty of the problem. To comply with the comment, we added mathematical representation of the target problem and analysis of its difficulty.

- **Validity of NVO game in various situations**: Reviewers eCZ8 and HRkt suggests additional experiments to verify extensibility/scalability to smaller $\epsilon$ values and larger dataset. We newly added these numerical results in the manuscript.

- **Error correction in the definition of pDP**: Reviewers xkkw and UXaT raised comments related to errors in definitions of pDP. We sincerely appreciate these kind comments and revised the manuscript.

These updates are temporarily highlighted for your convenience for checking.

We hope our response and revision sincerely satisfy all the reviewers.

Additionally, we confirmed that there was a typo in the title of the paper on openreview.net. Since the title could not be edited after submitting the paper, we will correct it later.

Thank you sincerely.

Best regards,

Authors.

---

### Meta-Review · Area_Chair_b8vE · 2023-12-05

**Metareview:**

The paper studies the problem of per-instance differential privacy, a weakening of the standard notion of differential privacy capturing the privacy of a specific individual with respect to a fixed dataset (as opposed to the privacy of all individuals with respect to all datasets). The paper seeks to use tools from game theory in order to add less noise while achieving privacy. Unfortunately, the paper is very badly written, with many mistakes (even in the definition of per-instance differential privacy) that were pointed out by the reviewers.
In its present form, it is not even easy to determine whether the paper is ultimately proposing a better algorithm for improving the restricted notion of per-instance differential privacy or the more standard general notion of differential privacy.

A significant re-write of the paper (in particular, clarifying what exactly are the privacy protection and utility properties being claimed) is needed before it is ready for publication.

**Justification For Why Not Higher Score:**

The paper is very badly written. A significant re-write of the paper is needed before it is ready for publication.

**Justification For Why Not Lower Score:**

N/A

---

### Decision · Program_Chairs · 2024-01-16

Reject